# Performative Reinforcement Learning in Gradually Shifting Environments

**Ben Rank**[♠1]      **Stelios Triantafyllou**[1]      **Debmalya Mandal**[♣◇2]      **Goran Radanovic**[♣1]

[1]Max Planck Institute for Software Systems (MPI-SWS)
[2]University of Warwick

## Abstract

When Reinforcement Learning (RL) agents are deployed in practice, they might impact their environment and change its dynamics. We propose a new framework to model this phenomenon, where the current environment depends on the deployed policy as well as its previous dynamics. This is a generalization of Performative RL (PRL) [Mandal et al., 2023]. Unlike PRL, our framework allows to model scenarios where the environment gradually adjusts to a deployed policy. We adapt two algorithms from the performative prediction literature to our setting and propose a novel algorithm called *Mixed Delayed Repeated Retraining* (MDRR). We provide conditions under which these algorithms converge and compare them using three metrics: number of retrainings, approximation guarantee, and number of samples per deployment. MDRR is the first algorithm in this setting which combines samples from multiple deployments in its training. This makes MDRR particularly suitable for scenarios where the environment's response strongly depends on its previous dynamics, which are common in practice. We experimentally compare the algorithms using a simulation-based testbed and our results show that MDRR converges significantly faster than previous approaches.

## 1 INTRODUCTION

When machine learning (ML) models are deployed in practice, they can affect the prediction target itself, causing a distribution shift. This problem has received significant attention in supervised learning and is termed as *performative*

---

♣Co-supervision.
◇Work done while at MPI-SWS.
♠benrank@mpi-sws.org

*prediction* [Perdomo et al., 2020]. In practice, it is often approached with *repeated retraining*: a practical solution for finding a (performatively) stable model, which does not suffer from further distribution shift.

Recently, Mandal et al. [2023] considered a reinforcement learning (RL) variant of this problem setting. In RL, performativity manifests itself as a shift in the environment, depending on the policy which was deployed by the learner. For example, the environment can model users of an online platform (e.g., recommender system or a chatbot), who adapt to the changes in the policy of the RL agent that controls the platform.

Mandal et al. [2023] formalizes this setting with a framework called *performative RL*, where the dynamics of a Markov decision process (MDP) $M_t$ depend on the current policy $\pi_t$. To find an approximately stable policy, they propose repeated retraining over the space of occupancy measures $\mathcal{C}(M_t)$ and the regularized objective

$$\max_{d \in \mathcal{C}(M_t)} \sum_{s,a} r_t(s,a) \cdot d(s,a) - \lambda \|d\|_2,$$

where $r_t$ is the reward function of $M_t$, the sum goes over all possible states $s$ and actions $a$, and $\lambda$ is a regularization factor.

However, this framework assumes that the environment only depends on the deployed policy and is independent of the previous environment. In many practical scenarios, this assumption does not hold. Going back to our examples from before, users are likely to manifest a learning behavior when interacting with the platform, and thus adapt their behavioral patterns gradually to any changes made in the platform, instead of adapting immediately after every change. Thus we consider an extension of the performative RL framework where the underlying MDP $M_t$ is gradually changing over time.

**Contributions** Following a similar line of work on performative prediction that considers gradual shifts in the

Table 1: **Overview of Our Results.** Convergence criteria for computing a $\delta$-approximate stable policy. $\lambda$ is a factor of the regularization, $\epsilon < 1$ indicates the dependence of the current environment on the previous environment, $\iota < 1$ indicates the dependence of the current environment on the deployed policy, $i$ denotes the current retraining iteration, $1 - p$ is the probability of achieving said approximate stable policy in the finite sample setting, $k$ denotes the number of repeated deployments of the same policy in MDRR, $|S|$ is the number of states, $|A|$ the number of actions, and $\gamma$ is the discount factor of the MDP.

| Algorithm | $\lambda$ | #retrainings | #samples per deployment |
|---|---|---|---|
| RR[exact] | $\mathcal{O}\left(\frac{|S|^{5/2}}{(1-\epsilon)(1-\gamma)^4}\right)$ | $\frac{\ln\left(\left(\frac{2}{1-\gamma}+(1+\sqrt{2})\sqrt{|S||A|}\right)/\delta\right)}{\ln(2/(1+\epsilon))}$ | N\A |
| DRR[exact] | $\mathcal{O}\left(\frac{\iota\cdot|S|^{5/2}}{(1-\epsilon)(1-\gamma)^4}\right)$ | $\ln\left(\left(\frac{2}{1-\gamma}\right)/\delta\right)$ | N\A |
| RR[fin] | $\mathcal{O}\left(\frac{\epsilon(|S|+\gamma|S|^{5/2})}{(1-\epsilon)(1-\gamma)^4}\right)$ | $\frac{\ln\left(\frac{\frac{2}{1-\gamma}+(1+\sqrt{2})\sqrt{|S||A|}}{\delta}\right)}{\ln(4/(3+\epsilon))}$ | $\tilde{\mathcal{O}}\left(\frac{|A|\psi}{\lambda^2(1-\epsilon)^4}\ln\left(\frac{i}{p}\right)\right)$[a] |
| DRR[fin] | $\mathcal{O}\left(\frac{\iota(|S|+\gamma|S|^{5/2})}{(1-\epsilon)(1-\gamma)^4}\right)$ | $\frac{\ln\left(\frac{2}{1-\gamma}/\delta\right)}{\ln(4/(3+\epsilon))}$ | $\tilde{\mathcal{O}}\left(\frac{|A|\psi}{\lambda^2}\ln\left(\frac{i}{p}\right)\right)$[a] |
| MDRR[fin] | as for DRR | as for DRR | $\frac{(v-1)v^{k-1}}{v^k-1}\tilde{\mathcal{O}}\left(\frac{|A|\psi}{\lambda^2}\ln\left(\frac{i}{p}\right)\right)$[a,b] |

[a] Here $\psi = \mathcal{O}\left(\frac{|S|^3\left(B+\sqrt{|A|}\right)^2}{\delta^4(1-\gamma)^6}\right)$ and we ignore all terms which are logarithmic in $|S|$, $|A|$ and $1/\delta$.

[b] $v > \frac{1}{\epsilon}$ is a hyperparamter of MDRR

[exact] results when the learner knows current environment $(P_t, r_t)$

[fin] results when the learner gets a finite set of samples from the current environment $(P_t, r_t)$

distribution [Brown et al., 2022, Li and Wai, 2022, Ray et al., 2022, Izzo et al., 2022], we model this scenario by assuming that the underlying MDP $M_t$ is dependent on both the deployed policy $\pi_t$ and the MDP from the previous round, i.e., $M_{t-1}$. Our overall goal is to analyze different repeated retraining approaches and provide characterization results that compare these approaches along the following three measures: a) attainable approximation quality (i.e., the minimum value of $\lambda$ for which the convergence is guaranteed), b) the number of retrainings which guarantees the convergence (signifying the compute needed to converge), and c) the sample complexity per deployment (signifying the number of data points that need to be collected). Our main contributions are as follows:

- *Framework:* An extension of the performative RL framework that can model gradual environment shifts, and an extension of the DRR algorithm from Brown et al. [2022], suitable for our framework.
- *Algorithm:* A novel repeated retraining algorithm, called MDRR, which compared to repeated retraining (RR) and DRR uses samples from multiple rounds of deployment, thereby reducing the number of samples needed per round.
- *Characterization results:* A characterization of three repeated retraining approaches: a canonical RR, DRR, and MDRR. Our analysis is a non-trivial combination of the proof techniques used by Mandal et al. [2023] and Brown et al. [2022] and brings additional insights about regular-

ization in performative RL. The overview of the results can be found in Table 1. At a high-level, our theoretical results suggest that DRR and MDRR fare better than RR in terms of the number of retrainings and sample complexity, as well as in terms of attainable approximation quality when the environment depends weakly on the current policy. When the environment depends strongly on the previous environment, MDRR fares better than RR and DRR in terms of samples per round. These results shed light on regularization in performative RL, and the importance of utilizing historic data to reduce it, thus obtaining better approximation quality.

- *Experiments:* Finally, we compare the algorithms in an experimental evaluation. In our experiments, MDRR outperforms RR and DRR in terms of the convergence speed and the quality of the solution obtained.

## 1.1 RELATED WORK

We relate our work to four lines of research: *Performative Prediction*, *Markov Games*, *Adversarial Markov Decision Processes*, and *Reinforcement Learning*. The latter two are discussed in Appendix A.

**Performative Prediction.** The study of performative prediction was initiated by Perdomo et al. [2020]. They investigate conditions under which repeated retraining converges to a performatively stable point. This study was

extended in various ways, including stochastic optimization [Mendler-Dünner et al., 2020], finding performatively optimal points [Miller et al., 2021, Izzo et al., 2021], multi-agent scenarios [Narang et al., 2023, Li et al., 2022] and using performativity to measure the power of firms [Hardt et al., 2022]. Mofakhami et al. [2023] use a different set of assumptions and provide convergence guarantees also in cases where the loss is not strongly convex in the parameters of the model. Most related to our setting are works that consider performative prediction under gradual shifts in the distribution [Brown et al., 2022, Li and Wai, 2022, Ray et al., 2022, Izzo et al., 2022], commonly known as *stateful* performative prediction [Brown et al., 2022]. All of the above works study performativity in supervised learning. In contrast, we consider reinforcement learning. However, we emphasize that some of our results are extensions of or inspired by those that appear in [Brown et al., 2022]. Most notably, we extend delayed repeated retraining, an algorithm proposed by [Brown et al., 2022], to our RL setting, and analyze its convergence guarantees. Furthermore, we introduce a novel algorithm inspired by delayed repeated retraining.

**Markov Games.** Our work is also related to the literature on stochastic or Markov games [Shapley, 1953] and multi-agent reinforcement learning [Zhang et al., 2021]. Much of the focus in multi-agent RL have been on computational and statistical aspects of learning Nash or correlated equilibria [Daskalakis et al., 2023, Wei et al., 2017, Bai et al., 2020, Jin et al., 2022]. Our setting is more related to multi-agent RL frameworks that consider Stackelberg or commitment policies [Letchford et al., 2012, Vorobeychik and Singh, 2012, Dimitrakakis et al., 2017, Zhong et al., 2021], where a principal agent commits a policy to which one or more followers best responds. Computing optimal commitment policies is in general computationally intractable [Letchford et al. [2012]. Hence, some restrictions on followers' response models are needed to enable computationally efficient learnability [Zhong et al. [2021]. Similarly, no-regret learning in a two-agent principal-follower setting where the follower independently learns or changes its policy over time is also in general computationally intractable [Radanovic et al., 2019, Bai et al., 2020]. However, if the dynamics of the follower's policy updates is not *adversarial*, tractable no-regret algorithms exist [Radanovic et al., 2019]. These restrictions on the follower are similar in spirit to the setting and the assumptions we consider in this paper, however, our setting is technically quite different: whereas these works focus on no-regret learning, we focus on performative RL and repeated retraining approaches.

## 2 PRELIMINARIES

We follow Perdomo et al. [2020], Brown et al. [2022] and Mandal et al. [2023] in defining the formal setting.

**Markov Decision Processes** We consider tabular Markov Decision Processes (MDPs), which consist of a finite state space $S$, finite action space $A$, discount factor $\gamma$ and initial state distribution $\rho$. We assume that the reward and transition probability functions change over time, as a response to the policy which the learner deploys. The learner deploys policy $\pi_t$ in round $t$ and the previous probability transition and reward function are $P_{t-1}$ and $r_{t-1}$. They then change to $P_t = \mathcal{P}(\pi_t, P_{t-1}, r_{t-1})$ and $r_t = \mathcal{R}(\pi_t, P_{t-1}, r_{t-1})$ respectively, according to the *response models* $\mathcal{P}$ and $\mathcal{R}$. Thus, the MDP in round $t$ is $M_t = (S, A, P_t, r_t, \rho)$.

When the learner deploys policy $\pi_{t+1}$, the probability of a trajectory $\tau = (s_k, a_k)_{k=0}^\infty$ to be realized in round $t$ is given by $\mathbb{P}_t^{\pi_t}(\tau) = \rho(s_0) \prod_{k=1}^\infty \pi_t(a_k|s_k) P_t(s_k, a_k, s_{k+1})$.

Given policy $\pi$ and initial state distribution $\rho$, we denote the value function at round $t$ as $V_t^\pi(\rho)$. It is defined as

$$V_t^\pi(\rho) = \mathbb{E}_{\tau \sim \mathbb{P}_t^\pi}\left[\sum_{k=0}^\infty \gamma^k r_t(s_k, a_k)|\rho\right].$$

The learner in round $t$ has access to the past MDPs $M_0, \ldots, M_{t-1}$, or a finite number of samples thereof.

**Solution Concept** We assume that when the learner deploys $\pi$ in every round, the MDP converges to the *limiting MDP* $M_\pi = (S, A, P_\pi, r_\pi, \rho)$, which is independent of the initial MDP. Using this, we can define the *performative value function* as

$$V_{\pi'}^\pi(\rho) = \lim_{t \to \infty} V_t^\pi(\rho|\pi_i = \pi' \,\forall i).$$

It is the value function of MDP $M_\pi$.

One common solution concept in this setting is to find a *performatively stable policy*, defined as follows.

**Definition 1** (Performatively Stable Policy). *We call a policy $\pi$ performatively stable, if it is the best response to the MDP $M_\pi$. That is, $\pi \in \arg\max_{\pi'} V_{\pi'}^\pi$.*

Given two performatively stable policices $\pi_1$ and $\pi_2$, their convex combination might not be performatively stable. Because of this, it is hard to use the standard formulation of RL. This problem is alleviated by using the linear programming formulation of RL. To describe this, we define the long term-state occupancy measure of a policy $\pi$ in MDP $M_t$ as $d_t^\pi(s, a) = \mathbb{E}_{\tau \sim \mathbb{P}_t^\pi}\left[\sum_{k=0}^\infty \gamma^k \mathbb{1}\{s_k = s, a_k = a\}\right]$. When given occupancy measure $d$, one can consider the following policy $\pi^d$, which has occupancy measure $d$.

$$\pi^d(a|s) = \begin{cases} \frac{d(s,a)}{\sum_b d(s,b)} & \text{if } \sum_a d(s,a) > 0 \\ \frac{1}{|A|} & \text{otherwise} \end{cases} \quad (1)$$

We consider that the learner parameterizes its policy by the occupancy measure and calculates the policy via (1).

In an unregularized setting, we would say that a occupancy measure $d_S$ is performatively stable if it is the optimal solution to the following linear program.

$$d_S^* \in \arg\max_{d \geq 0} \sum_{s,a} d(s,a) r_{d_S^*}(s,a) \qquad (2)$$

$$\text{s.t.} \sum_a d(s,a) = \rho(s) + \gamma \cdot \sum_{s',a} d(s',a) P_{d_S^*}(s',a,s) \; \forall s$$

where we denote $P_d = P_{\pi_d}$ and $r_d = r_{\pi_d}$. This describes an occupancy measure which is itself the best response against the current MDP.

But, similar to prior work, to make the theoretical analysis feasible, we assume the following regularized version of optimization problem (2). A stable occupancy measure $d_S$ is defined by

$$d_S \in \arg\max_{d \geq 0} \sum_{s,a} d(s,a) r_d(s,a) - \frac{\lambda}{2} \|d\|_2^2 \qquad (3)$$

$$\text{s.t.} \sum_a d(s,a) = \rho(s) + \gamma \cdot \sum_{s',a} d(s',a) P_d(s',a,s) \; \forall s.$$

Here $\lambda$ is a constant regularization factor which describes the strong-concavity of the objective. This describes an occupancy measure which is itself the best response against a regularized objective of the current MDP. If a learner updates their occupancy measure using the best response against a $L2$-regularized objective, (3) describes an occupancy measure which would not change under such an update, i.e. be stable.

In our results, we provide lower bounds for how small $\lambda$ can be to guarantee convergence. Furthermore, in Appendix C.3 we show that (3) approximates the unregularized objective (2).

**Sensitivity Assumption** We overload the notation to write the response models in the following form. For every occupancy measure $d$, let $\mathcal{P}(d, P, r) = \mathcal{P}(\pi_d, P, r)$ and $\mathcal{R}(d, P, r) = \mathcal{R}(\pi_d, P, r)$.

For the learner to make use of this past information, we use the following sensitivity assumption, which are commonly used in performative prediction.

**Assumption 1** (sensitivity). *Consider some* $\iota_p, \iota_r, \epsilon_{p,p}, \epsilon_{p,r}, \epsilon_{r,p}, \epsilon_{r,r} \geq 0$ *with* $\iota = \iota_p + \iota_r < 1$, $\epsilon_p = \epsilon_{p,p} + \epsilon_{r,p} < 1$ *and* $\epsilon_r = \epsilon_{p,r} + \epsilon_{r,r} < 1$. *Assume*

$$\|\mathcal{P}(d, P, r) - \mathcal{P}(d', P', r')\|_2$$
$$\leq \iota_p \|d - d'\|_2 + \epsilon_{p,p} \|P - P'\|_2 + \epsilon_{p,r} \|r - r'\|_2 \text{ and}$$
$$\|\mathcal{R}(d, P, r) - \mathcal{R}(d', P', r')\|_2$$
$$\leq \iota_r \|d - d'\|_2 + \epsilon_{r,p} \|P - P'\|_2 + \epsilon_{r,r} \|r - r'\|_2$$

*for any occupancy measures* $d, d'$, *reward functions* $r, r'$ *and probability transition functions* $P, P'$.

Assumption 1 ensures that when the learner deploys a new policy, the new MDP does not drift too far from the old MDP. In Appendix C.1 we discuss one example where this assumption commonly holds.

When $\epsilon_p, \epsilon_r < 1$, the mapping from $(P, r)$ to $(\mathcal{P}(d, P, r), \mathcal{R}(d, P, r))$ is a contraction for any occupancy measure $d$ (Proof in Appendix C.4). Therefore, if the learner deploys the same policy $\pi$ in every round, $P_t$ and $r_t$ asymptotically converge to some $P_\pi$ and $r_\pi$ respectively and we don't need to assume this explicitly.

To simplify the exposition of the results in the main paper, we assume that the following assumption holds, without explicitly stating it in the results.

**Assumption 2.** *For the results in the main part of the paper, we assume that* $\epsilon_{p,p} = \epsilon_{p,r} = \epsilon_{r,p} = \epsilon_{r,r} = \frac{\epsilon}{2}$, $\iota_p, \iota_r \leq \frac{\epsilon}{2}$ *for some* $\epsilon < 1$ *and* $\frac{9\gamma|S|}{(1-\gamma)^2} \geq 1$.

Assumption 2 is not critical – as we show in the appendix, our results easily generalize when we do not assume it.

**Sample Generation Model** We also consider finite-sample versions of the algorithms we propose. For this we use the following sample generation model. In round $t$, let $\overline{d}_t$ be the occupancy measure of $\pi_t$ under dynamics $P_t$. Note that this is different than the occupancy measure which the learner uses to calculate its policy $\pi_{d_t}$, since this was calculated using different dynamics. We then define the normalized occupancy measure $\tilde{d}_t(s,a) = (1-\gamma)\overline{d}_t(s,a)$. Each sample in round $t$ is a tuple $(s, a, r, s')$ and is generated in the following way. First a state, action pair is sampled i.i.d. according to $(s,a) \sim \tilde{d}_t$, then reward as $r = r_t(s,a)$ and then the next state $s' \sim P_t(\cdot|s,a)$. This is a standard model of sample generation in offline RL [Munos and Szepesvári, 2008, Farahmand et al., 2010, Xie and Jiang, 2021, Mandal et al., 2023].

## 3 REPEATED RETRAINING (RR)

One common approach in performative prediction is *repeated retraining (RR)*, where the learner updates its policy at every round, by best responding to the current environment. In this section, we explore guarantees for when this approach converges to a stable occupancy measure.

In RR we assume that the learner updates its policy every round in such a way that it is optimal for the regularized objective of the current MDP $M_t$. In particular, we define $d_{t+1}$ to be a solution to the following optimization problem.

$$\max_{d \geq 0} \sum_{s,a} d(s,a) r_t(s,a) - \frac{\lambda}{2} \|d\|_2^2 \qquad (4)$$

$$\text{s.t.} \sum_a d(s,a) = \rho(s) + \gamma \cdot \sum_{s',a} d(s',a) P_t(s',a,s) \; \forall s$$

We go on to show that RR converges to a stable occupancy measure.

**Theorem 1** (informal, details in Appendix D.2). *Assume that Assumption 1 holds and $\lambda = \mathcal{O}\left(\frac{|S|^{5/2}}{(1-\epsilon)(1-\gamma)^4}\right)$. Then for any $\delta > 0$ we have,*

$$\|d_t - d_S\|_2 \leq \delta,$$

$$\text{for all } t \geq \frac{\ln\left(\left(\frac{2}{1-\gamma} + \left(1+\sqrt{2}\right)\sqrt{|S||A|}\right)/\delta\right)}{\ln(2/(1+\epsilon))} .$$

The bound on $\lambda$ in Theorem 1 is comparable to the one required in standard Performative RL, there are only differences in the constants and the $\epsilon$ factors. The bound on the number of rounds $t$ in standard Performative RL is $2\ln\left(\frac{2}{\delta(1-\gamma)}\right) / \left(1 - \frac{|S|^{5/2}\epsilon}{\lambda(1-\gamma)^4}\right)$, which is comparable to the bound here, when only considering the $\delta$ parameter. We note that the bound on $t$ in Theorem 1 does depend on $\lambda$, but for the simplicity of the exposition it is swapped by the lower bound on $\lambda$ instead. The full theorem is found in Appendix D.2.

The proofs of this paper are found in the appendix. In general, the proofs rely on a non-trivial combination of adapting arguments from Brown et al. [2022] to the RL setting and using results from Mandal et al. [2023]. Additionally, we extend the analysis by introducing a distinction between the parameter $\iota$ indicating how the environment adapts to a deployed policy, and $\epsilon$ indicating how strongly the environment depends on the previous environment. We therefore view our main contribution in this section and Section 4 as bridging the gap between the theoretical findings of Mandal et al. [2023] and the often more realistic assumptions made by history-dependence, as in Brown et al. [2022]. In section 5 we will introduce a novel algorithm.

### 3.1 FINITE SAMPLE GUARANTEES

Theorem 1 assumes that the learner knows the exact environment when updating its policy. In practice, this is usually too strong of an assumption, since the learner typically has access to only a finite number of samples drawn via the deployed policy on the adopted environment. In this subsection, we first discuss some general considerations for this new setting and then show that RR also converges here.

**Update rule for RR** The learner has access to i.i.d. drawn set of samples $F_t$ for each round $t$. In round $t$, let $m_t := |F_t|$ be the number of samples.

As prior work, we use the following *empirical Lagrangian* to devise an optimization problem in the finite sample set-

ting [Mandal et al., 2023].

$$\hat{\mathcal{L}}(d, h; t) = -\frac{\lambda}{2}\|d\|_2^2 + \sum_s h(s)\rho(s)$$

$$+ \sum_{(s,a,r,s')\in F_t} \frac{d(s,a)}{\bar{d}_t(s,a)} \cdot \frac{r - h(s) + \gamma h(s')}{m_t(1-\gamma)} \quad (5)$$

Here $h$ is the Lagrange multiplier with one entry for each $s \in S$.

The empirical Lagrangian is defined in such a way that when we take its expectation over samples, we obtain the exact Lagrangian $\mathcal{L}$ of optimization (4). One can show that the empirical Lagrangian $\hat{\mathcal{L}}$ lies in a neighborhood of the true Lagrangian $\mathcal{L}$ almost certainly. The learner repeatedly solves

$$(d_{t+1}, h_{t+1}) = \arg\max_d \ \arg\min_h \hat{\mathcal{L}}(d, h; t) . \quad (6)$$

We need a further assumption, which ensures an overlap in the occupancy measure between the behavioral policy and the target policy space. This assumption is standard in offline RL [Munos and Szepesvári, 2008, Zhan et al., 2022, Mandal et al., 2023]. Without such an overlap, it is unclear how the learner would compute an optimal policy.

**Assumption 3.** *Assume we are given an integer $k$. Given occupancy measure $d$, initial transition probability function $P_0$ and initial reward function $r_0$, let $P_t$ and $r_t$ be the result after the learner deploys $\pi^d$ for $t$ rounds. Let $d_t^*$ be the solution to optimization problem (4). Let $\bar{d}_t$ be the occupancy measure of $\pi^d$ in $P_t$. Then there exists $B > 0$ such that for all $d$ and $t \leq k$ it holds that*

$$\max_{s,a}\left|\frac{d_k^*(s,a)}{\bar{d}_t(s,a)}\right| \leq B .$$

Note that we only need overlap for state-action pairs where the optimal policy $d_t^*$ is non-zero. So values where $d_t^*(s,a)$ is 0 are allowed iff $\bar{d}_t(s,a)$ is 0 for all $t \leq k$.

We can then show the following guarantee for RR.

**Theorem 2** (informal, details in Appendix D.3). *Suppose that overlap Assumption 3 holds for $k = 1$ with parameter $B$ and Assumption 1 holds. Let $p > 0$. Then for $\lambda = \mathcal{O}\left(\frac{\epsilon(|S|+\gamma|S|^{5/2})}{(1-\epsilon)(1-\gamma)^4}\right)$, $m_t = \tilde{\mathcal{O}}\left(\frac{|A||S|^3\left(B+\sqrt{|A|}\right)^2}{\delta^4(1-\gamma)^6\lambda^2(1-\epsilon)^4}\ln\left(\frac{t}{p}\right)\right)$[1] and for any $\delta > 0$, with probability at least $1 - p$,*

$$\|d_t - d_S\|_2 \leq \delta \text{ for all } t \geq \frac{\ln\left(\frac{\frac{2}{1-\gamma} + \left(1+\sqrt{2}\right)\sqrt{|S||A|}}{\delta}\right)}{\ln\left(4/\left(3+\epsilon\right)\right)} + 1 .$$

---

[1] Here we ignore all terms which are logarithmic in $|S|$, $|A|$ and $1/\delta$

The bounds here are similar to the bounds in standard Performative RL. For $\lambda$, there is no $\gamma$ in the numerator and the $\epsilon$ parameters are a bit different in the standard setting. The number of retrainings also has a factor of $\ln(1/((1-\gamma)\delta))$ in standard Performative RL.

# 4 DELAYED REPEATED RETRAINING

A different approach inspired by work from Brown et al. [2022], is to not update the policy every round, but wait a number of $k$ rounds before each update. Then the policy is updated using only the environment from the last round of the $k$ deployments. Algorithm 1 illustrates this approach, called *Delayed Repeated Retraining* (DRR).

The advantage of DRR is that during the rounds of repeatedly deploying the same policy, the MDP can somewhat stabilize and the learner might need a lower amount of retrainings and therefore less compute.

For the result, we use the following definition.

**Definition 2.** *Let* $\mathrm{d}_{P,r}$ *be the maximal distance between any environment and its successive environment, i.e.*

$$\mathrm{d}_{P,r} := \max_{P,r,d}\left(\|\mathcal{P}(P,r,d) - P\|_2 + \|\mathcal{R}(P,r,d) - r\|_2\right).$$

**Theorem 3** (informal, details in Appendix E.1). *Let* $d_i$ *be computed by DRR with* $k = \ln^{-1}\left(\frac{1}{\epsilon}\right)\ln\left(\frac{\mathrm{d}_{P,r}}{\delta\iota}\right)$. *Suppose Assumption 1 holds and* $\lambda = \mathcal{O}\left(\frac{\iota\cdot|S|^{5/2}}{(1-\epsilon)(1-\gamma)^4}\right)$. *Then for any* $\delta > 0$, *we have*

$$\|d_i - d_S\|_2 \leq \delta \quad \text{for all } i \geq \ln\left(\left(\frac{2}{1-\gamma}\right)/\delta\right).$$

The regularization parameter $\lambda$ has an $\iota$ factor in DRR, but not in RR (see Theorem 1). The factor $\iota$ is close to 0, if the MDP does not react strongly to the current policy. In such settings, the conditions for $\lambda$ in DRR are substantially relaxed. In addition, the number of retrainings required for DRR is much smaller than for RR. However, RR may require fewer total rounds than DRR.

---

**Algorithm 1:** Delayed Repeated Retraining
1: **Input:** radius $\delta$, initial transition probability $P_0$ and reward function $r_0$, initial occupancy measure $d_0$, number of deployments $k$
2: **for** $i = 0, 1, 2, \ldots$ **do**
3:     **for** $g = 1, \ldots, k$ **do**
4:         // deploy $\pi_{d_i}$:
5:         $P_{i\cdot k+g} \leftarrow \mathcal{P}(d_i, P_{i\cdot k+g-1}, r_{i\cdot k+g-1})$
6:         $r_{i\cdot k+g} \leftarrow \mathcal{R}(d_i, P_{i\cdot k+g-1}, r_{i\cdot k+g-1})$
7:     Update policy to $\pi_{d_{i+1}}$

---

## 4.1 FINITE SAMPLE GUARANTEES

In DRR with finite samples, the learner again applies the same policy for several rounds. After that it updates its policy using samples drawn from the most recent environment. For this, the learner uses optimization problem (6).

**Theorem 4** (informal, details in Appendix E.2). *Let* $d_i$ *be computed by finite sample DRR with* $k = \ln^{-1}\left(\frac{1}{\epsilon}\right)\ln\left(\frac{5\cdot\mathrm{d}_{P,r}}{\delta\iota}\right)$. *Suppose the Assumption 1 holds and Assumption 3 holds for* $k$ *and parameter* $B$. *Let* $p > 0$. *Furthermore assume* $\lambda = \mathcal{O}\left(\frac{\iota(|S|+\gamma|S|^{5/2})}{(1-\epsilon)(1-\gamma)^4}\right)$. *Then for*

$$m_i = \tilde{\mathcal{O}}\left(\frac{|A||S|^3\left(B+\sqrt{|A|}\right)^2}{\delta^4(1-\gamma)^6\lambda^2}\ln\left(\frac{i+1}{p}\right)\right)[1], \text{ and any } \delta > 0,$$

*with probability at least* $1 - p$,

$$\|d_i - d_S\|_2 \leq \delta \quad \text{for all } i \geq \frac{\ln\left(\frac{2}{1-\gamma}/\delta\right)}{\ln\left(4/\left(3+\epsilon\right)\right)} + 1.$$

In this result, $\lambda$ has a factor of $\iota$, whereas RR has a factor of $\epsilon$ (see Theorem 2). In prior work, the difference of $\epsilon$ and $\iota$ was ignored and the two were assumed to be the same [Brown et al., 2022]. As we see here however, interesting properties emerge when we explicitly assume that they are not the same. In settings where the environment does not respond strongly to the current policy, but strongly depends on the previous environment, $\epsilon$ is larger than $\iota$, substantially relaxing the conditions on $\lambda$ for DRR. DRR also requires less samples, by a factor of $(1-\epsilon)^4$ when assuming equal $\lambda$. The number of retrainings also is less for DRR. Still, RR may need fewer rounds of retraining overall because DRR only retrains every $k$th round. Assumption 3 is stricter for DRR, because it has a larger $k$-parameter than RR. The $k$ parameter in Assumption 3 indicates how far into future rounds the overlap of occupancy measures has to reach.

# 5 MIXED DELAYED REPEATED RETRAINING (MDRR)

Consider a scenario where in each round the learner gets a limited number of samples from the MDP. In this scenario, in each training step DRR would use samples from one round only. But using samples from multiple rounds would allow the learner to use more samples overall, reducing variance and potentially improving convergence.

However, it is challenging to determine how the learner should combine samples from multiple rounds. Should they optimize using all available samples collectively, or should they use more samples from recent rounds and less from older ones? Additionally, it is uncertain whether such a method would converge and, if so, whether it would offer

any benefits. To address these questions, we present a novel algorithm that:

- Uses samples from multiple rounds.
- Allows for prioritizing recent samples while still incorporating older ones.
- If the response of the environment depends strongly on the previous MDP, achieves convergence with fewer samples per round. If additionally the number of provided samples per deployment is low, it provides better approximation guarantees.

The algorithm uses a new optimization problem, which can be viewed as an extension of the previous empirical Lagrangian (5) to multiple rounds:

$$
\hat{\mathcal{L}}^M(d, h, i) = -\frac{\lambda}{2}\|d\|_2^2 + \sum_s h(s)\rho(s)
$$
$$
+ \sum_{g=1}^{k} \sum_{\substack{(s,a,r,s') \\ \in F_{i\cdot k+g}}} \frac{1}{U_i} \frac{d(s,a)}{\overline{d}_{i\cdot k+g}(s,a)} \frac{r - h(s) + \gamma h(s')}{1 - \gamma} \quad (7)
$$

Here we define by $\overline{d}_{i\cdot k+g}$ the occupancy measure of policy $\pi_{d_i}$ under dynamics $P_{i\cdot k+g}$. $U_i$ denotes the total number of samples, i.e. $U_i := \sum_{g=1}^{k} |F_{i\cdot k+g}|$. The learner thus optimizes over samples from multiple rounds of deployment.

But there is an inherent trade-off: recent samples contain more information about the current environment, but using earlier samples allows the total set of samples to be larger.

To balance this trade-off, the approach here is to use more samples from recent rounds and less samples from early rounds. For illustration, let's assume that the learner didn't update its policy since MDP $M_{i\cdot k} = (S, A, P_{i\cdot k}, r_{i\cdot k}, \rho)$ and updates every $k$ rounds. Then they might take $m$ samples from $M_{i\cdot k+1}$, $mv$ samples from $M_{i\cdot k+2}$ (for $v > 1$), $mv^2$ samples from $M_{i\cdot k+3}, \ldots,$ and $mv^{k-1}$ samples from $M_{i\cdot k+k}$. If $v$ is close to 1, the learner takes approximately equal number of samples from all rounds. If $v$ is large, and $m$ small, the learner focuses more on recent rounds. The pseudocode for this approach is shown in Algorithm 2, we call it *Mixed Delayed Repeated Retraining* (MDRR).

In MDRR the learner uses $m_{i\cdot k+g} = \frac{v-1}{v^k-1}v^{g-1}U_i$ samples from environment $(P_{i\cdot k+g}, r_{i\cdot k+g})$ (for each $g = 1, \ldots, k$), where $U_i$ denotes the total number of samples used to compute $d_{i+1}$.

**Theorem 5** (informal, details in Appendix F.2). *Let $d_i$ be computed by MDRR with $k \geq \frac{\ln\left(\frac{\epsilon(v-1)}{v\epsilon-1}\right) + \ln\left(\frac{5(1-\epsilon)\,\mathrm{d}_{P,r}}{\iota\delta}\right)}{\ln(1/\epsilon)}$. Suppose the Assumption 1 holds and the overlap Assumption 3 holds for $k$ and parameter $B$. Let $p > 0$. Also assume that $\lambda = \mathcal{O}\left(\frac{\iota(|S|+\gamma|S|^{5/2})}{(1-\epsilon)(1-\gamma)^4}\right)$. Further let $U_i = \tilde{\mathcal{O}}\left(\frac{|A||S|^3\left(B+\sqrt{|A|}\right)^2}{\delta^4(1-\gamma)^6\lambda^2}\ln\left(\frac{i+1}{p}\right)\right)$[1] be the total number of*

---

**Algorithm 2:** Mixed DRR (MDRR)

1: **Input:** radius $\delta$, initial $P_0$ and $r_0$, initial occupancy measure $d_0$, hyperparameters $v$ and $k$, total number of samples for each round $U_i$
2: **for** $i = 0, 1, 2, \ldots$ **do**
3:      **for** $g = 1, \ldots, k$ **do**
4:          $P_{i\cdot k+g} \leftarrow \mathcal{P}(d_i, P_{i\cdot k+g-1}, r_{i\cdot k+g-1})$
5:          $r_{i\cdot k+g} \leftarrow \mathcal{R}(d_i, P_{i\cdot k+g-1}, r_{i\cdot k+g-1})$
6:          $F_{i\cdot k+g} \leftarrow$ draw $\frac{v-1}{v^k-1}v^{g-1}U_i$ samples from $(P_{i\cdot k+g}, r_{i\cdot k+g})$
7:      Update occupancy measure $d_{i+1} \leftarrow \arg\max_d \min_h \hat{\mathcal{L}}^M(d, h, i)$

---

samples in retraining-round $i$ and $v > \frac{1}{\epsilon}$. Then for any $\delta > 0$, with probability at least $1 - p$,

$$
\|d_i - d_S\|_2 \leq \delta \quad \text{for all } i \geq \frac{\ln\left(\frac{2}{1-\gamma}/\delta\right)}{\ln\left(4/(3+\epsilon)\right)} + 1 .
$$

The proof of this result involves showing that the empirical Lagrangian (7) approximates an exact Lagrangian of the optimization problem where the MDP is a mixture of MDPs from different rounds. We then show that the solution to this optimization problem approximates the solution of an exact one-step update with the limiting MDP (i.e. the MDP which the environment converges to if the learner repeatedly applies the current policy). In a last step we apply arguments similar to the proof of convergence for DRR.

To compare MDRR to RR and DRR, let's first consider the case when $\epsilon$ is close to 1. This holds when the environment responds strongly to the old environment, for example when the new environment after one step is a slight alteration of the old environment. We expect this property to hold in many applications, because the environment shift typically happens only slowly over time. We anticipate that MDRR performs particularly well in those settings, because it uses samples from old environments, and if those environments are close to the current environment, those samples are more informative. And indeed, this is what we observe. The number of samples required in line 6 of MDRR is smaller by a factor of $\frac{v^k-v^{k-1}}{v^k-1}$, which converges to $(v-1)/v$ for large $k$. When $\epsilon$ is close to 1, we can set $v$ close to 1, resulting in a significant decrease in the required number of samples.

The regularization parameter $\lambda$ is the same as for DRR and has a factor of $\iota$ compared to RR which has a factor of $\epsilon$. But note that the number of samples has a factor of $1/\lambda^2$ in all three algorithms, therefore in settings where there are few samples, one needs larger $\lambda$ to guarantee convergence. However, because MDRR requires less samples per round than RR and DRR in those settings, it requires smaller values of $\lambda$. The number of retrainings is similar to DRR and significantly less than for RR.

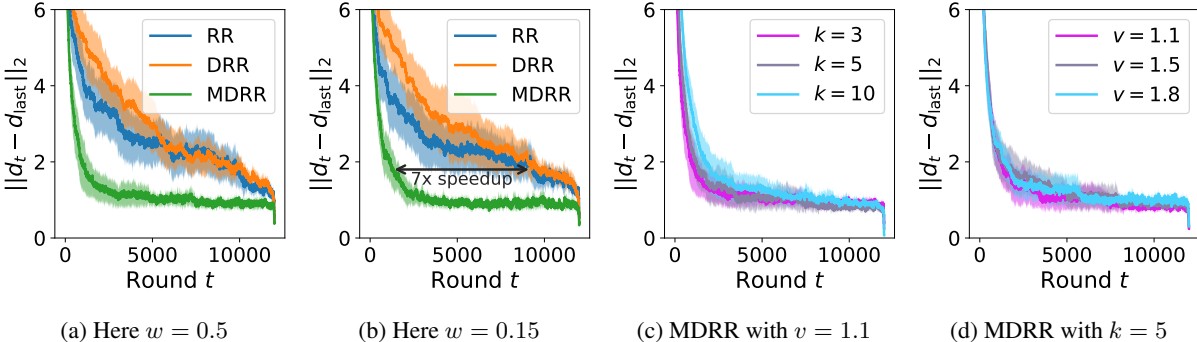

|     | (a) Here $w = 0.5$ | (b) Here $w = 0.15$ | (c) MDRR with $v = 1.1$ | (d) MDRR with $k = 5$ |

Figure 1: The figures show the distance of the current occupancy measure from the average of the last 10 in that run (after 11990 deployments). The data represent means computed over 20 trials, along with their 95% confidence intervals. Unless otherwise noted, the settings are $k = 3$ for DRR and MDRR, $v = 1.1$ for MDRR, 1000 trajectories per iteration, $B = 10$, $\lambda = 0.1$ and $w = 0.5$ Figure 1a and 1b compare the three algorithms to one another, while Figure 1c compares MDRR with different values for the hyperparameter $k$ and Figure 1d compares MDRR with different values for the hyperparameter $v$.

In general, we see that MDRR performs particularly well in settings where the environment responds strongly to the previous environment in a given round, which likely is a scenario often present in practice.

## 6 EXPERIMENTS

**Environment** In order to compare the three algorithms in a fair and tractable experimental setup, we use a variation of the experimental testbed from Mandal et al. [2023], with two agents controlling an actor in a grid-world. In our testbed, agent $A_1$ proposes a control policy for the actor and $A_2$ responds by overriding some of the actions taken by the control policy. Hence, $A_1$'s effective environment is performative. More information about this experimental setup can be found Appendix B.1.

To simulate a slow response, $A_2$ plays a weighted combination of its last policy and a softmax of its optimal $Q$-values. Specifically, the policy of $A_2$ in round $i$ is

$$\pi_i^2(a|s) = w \cdot \frac{e^{Q_2^{*|\pi_1}(s,a)}}{\sum_{a' \in A} e^{Q_2^{*|\pi_1}(s,a')}} + (1-w) \cdot \pi_{i-1}^2(a|s) \quad (8)$$

Here $Q_2^{*|\pi_1}(s, a)$ are the optimal $Q$-values for $A_2$, while $w$ describes the responsiveness of the environment towards the deployed policy of $A_1$. For small $w$, the environment responds strongly to the current policy, while for large $w$ the environment is less responsive to the current policy.

**Implementation** We study the finite sample setting, and sample trajectories instead of taking single samples from occupancy measures. The learner solves the min-max-problem (6) using a follow-the-regularized-leader algorithm described in Appendix B.3. To evaluate the speed at which the algorithms reach a stable occupancy measure, we evaluate how the occupancy measure at each round compares

to the average of the last 10 occupancy measures, which we denote by $d_{\text{last}}$.[2]

**Performance** In Figures 1a and 1b we see that MDRR converges the fastest to $d_{\text{last}}$. This is true both for the setting where the environment changes faster ($w = 0.5$, Figure 1a) and when it changes more slowly ($w = 0.15$, Figure 1b). This is the case even though MDRR uses less retrainings than RR. But MDRR uses more samples per retraining, and this seems to lead to better convergence properties in the exposed settings. This also means that MDRR has lower variance, as indicated by the smaller confidence intervals. In Appendix B.6 we additionally study settings with larger values of $w$, where the environment is more dynamic. Also here MDRR significantly outperforms RR and DRR.

**Choice of Hyperparameters** As we can see in Figure 1c, the convergence properties of MDRR for different values of $k$ are similar. As we can see in Figure 1d, in the range of $v = 1.1$ to $v = 1.8$, there does not seem to be much difference in speed of convergence. The results indicate that MDRR is robust to the choice of its hyperparameters.

**Compute details** The experiments in Figure 1 were conducted on a compute cluster with each machine having 4 Intel Xeon E7-8857 v2 CPUs and 1.5 TB of RAM. It took approximately 80 to 100 hours per algorithm to complete each experiment.

## 7 CONCLUSION

This work initiates the study of performative RL in scenarios where the environment changes gradually. We introduce

---

[2]Code available at https://github.com/rank-and-files/performative-rl-gradually-shifting-envs

different algorithms in this setting and compare them extensively both theoretically and experimentally. Our results suggest that our novel MDRR algorithm performs particularly well in this setting, and it would be interesting to investigate similar algorithms in performative prediction.

## Acknowledgements

This research was, in part, funded by the Deutsche Forschungsgemeinschaft (DFG, German Research Foundation) – project number 467367360.

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

# Performative Reinforcement Learning in Gradually Shifting Environments
## (Supplementary Material)

**Ben Rank**[♠1]    **Stelios Triantafyllou**[1]    **Debmalya Mandal**[♣◇2]    **Goran Radanovic**[♣1]

[1]Max Planck Institute for Software Systems (MPI-SWS)
[2]University of Warwick

## Table of Contents

## A  ADDITIONAL RELATED WORK

In this section we present some more related work on Adversarial MDPs and Reinforcement Learning (RL).

**Adversarial MDPs.** More broadly, our framework is related to the literature on adversarial and non-stationary MDPs, which extensively studied online learning under adversarial and non-stationary rewards and transitions[Even-Dar et al., 2004, 2009,

Abbasi Yadkori et al., 2013, Yu and Mannor, 2009, Rosenberg and Mansour, 2019, Cheung et al., 2020, Wei and Luo, 2021]. The positive results therein, in particular, no-regret guarantees when both rewards and transitions evolve over time, often assume budget constraints on how many times and by how much the underlying MDP model can change [Abbasi Yadkori et al., 2013, Cheung et al., 2020, Wei and Luo, 2021]. We instead rely on sensitivity assumptions (Assumption 1), introduced in Section 2.

**Reinforcement Learning.** We also mention the recent work on RL in Newcomb-like environments [Bell et al., 2021], whose framework is similar to the original performative RL framework of Mandal et al. [2023]. There, the focus is on the convergence of value-based RL algorithms; we focus on repeated retraining and allow the environment response model to gradually change over time. From a practical point of view, repeated retraining is similar to alternating optimization for game-theoretic bi-level optimization problems in RL(e.g., [Rajeswaran et al., 2020, Mohammadi et al., 2023]). The latter can be thought of as a training framework for finding optimal commitment policies in Markov games, whereas the former repeatedly deploys a policy, collects data, and trains a new policy using offline RL. In that regard, we also relate this paper to the vast literature on offline RL [Levine et al., 2020]. From a technical point of view, the most relevant aspects are coverage assumptions and data generation process: we consider the ones from [Mandal et al., 2023], which are based on [Zhan et al., 2022, Munos and Szepesvári, 2008].

# B ADDITIONAL EXPERIMENTAL DETAILS

This section discusses more details on the experiments. Subsection B.1 explains the environment further, subsections B.2 and B.3 discuss further algorithmic details, subsection B.4 discusses the type and amount of compute used, and in subsection B.5 we sanity-check if the comparison presented in the main paper is fair.

## B.1 EXPLANATION OF THE ENVIRONMENT

The experimental setting is an adapted version of the one from Mandal et al. [2023]. We consider the grid world environment depicted in figure 2. There is one actor in this grid-world environment, which is controlled by two agents, agent $A_1$ and agent $A_2$. The actor starts randomly in one of the $S$ states, with uniform probability. $A_1$ can decide where the actor goes by choosing one of the directions left, right, up or down. $A_2$ can decide to intervene on the direction which $A_1$ chose. The actions of $A_2$ are not-intervene, left, right, up or down. In case $A_2$ chooses not-intervene, the direction chosen by $A_1$ is used. Otherwise, the direction chosen by $A_2$ gets used.

Both agents are reinforcement learners with different goals. $A_1$ optimizes according to the grid-world in figure 2. $A_2$ optimizes according to a perturbed grid-world, where each blank, $F$ or $H$ cell is the same as for $A_1$ with probability 0.7. With probability 0.3, it gets changed to either blank, $F$ or $H$ (chosen uniformely at random).

$A_1$ and $A_2$ get a negative reward of $-0.01$ if the actor visits a blank or an $S$ cell, a slightly increased negative reward of $-0.02$ if visiting a $F$ cell and a large negative reward of $-0.5$ for $H$ cells. Additionally, when $A_2$ decides to intervene, an additional cost of $-0.05$ is inflicted on it.

$A_1$ is the main learner which performs RR, DRR or MDRR. $A_2$ models the response of the environment.

$A_2$ starts by playing the policy which does never intervene. In each iteration, first $A_1$ optimizes its policy, and then $A_2$ responds to the policy played by $A_1$. $A_2$ slowly adapts to the current played policy by agent 1 in each round, by using a mixture between the last played policy of $A_2$ and the softmax over the current optimal $Q$-values, as described in equation (8) in the main paper.

Furthermore, we use $\gamma = 0.9$ for both $A_1$ and $A_2$ and a maximum trajectory length of 50, i.e. after 50 steps, the trajectory is

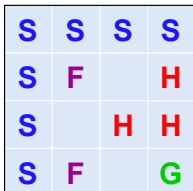

Figure 2: The grid-world.

cut off. Instead of using the exact occupancy measures $\bar{d}_i$ in the optimization, we approximate them using the trajectories.

## B.2 COMPUTING SAMPLE LISTS FOR MDRR

In this subsection we describe a practical way to compute samples from MDRR.

Recall that MDRR uses $m_{ik+t} = w_t U_i$ samples in iteration $i$ from round $t$, where $w_t = \frac{(v-1)v^{t-1}}{v^k-1}$ and $U_i$ is the total number of samples used in for the $i$-th retraining. In practice, we assume that the learner is given some samples for each round.

In practice, we use a slightly different algorithm to compute the number of samples MDRR uses, because of two reasons. The first reason is that $w_t U_i$ could be non-integral. The second reason is that even though MDRR needs $m_t$ samples in round $t$, samples from rounds after $t$ could also count towards this, if the same policy was applied in the rounds between. This is because those samples are collected after more repeated applications of the same policy. Therefore the environment at this point is closer to the limiting environment than in round $t$ and using additional samples from higher rounds would increase performance more than using samples from round $t$.

To calculate the number of samples MDRR uses from each round, we propose Algorithm 3, which we explain in the following. In the following we use the terms *list* and *sequence* somewhat loosely to refer to linked lists of samples and linked lists of linked lists of samples respectively.

Algorithm 3 takes as input a sequence of lists of samples $S_1, \ldots, S_k$ and weights $w_1, \ldots, w_k \in \mathbb{R}^+$. We can think of $S_1$ to be the number of samples in step $ik + 1$ for some $i$, $S_2$ to be the number of samples in step $ik + 2$, etc. . Algorithm 3 fulfills the following property.

**Theorem 6.** *Algorithm 3 outputs a sequence $\mathcal{F} = [F_1, \ldots, F_k]$, which contains the maximal number of samples $|\mathcal{F}|$ such that*

1. $F_t \subseteq S_t$ for all $t \in \{1, \ldots, k\}$ and
2. $|[F_t, \ldots, F_k]| \geq \sum_{t'=t}^{k} w_{t'} |\mathcal{F}|$ for all $t \in \{1, \ldots, k\}$ .

*where we denote by $|[F_t, \ldots, F_k]|$ the number of samples in total in $F_t, \ldots, F_k$. Similarly $|\mathcal{F}|$ is the number of samples in $\mathcal{F}$.*

Item 1 guarantees that $F_t$ only contains samples from $S_t$. Item 2 guarantees that for each round $t$, there is a sufficient number of samples assigned to this step either by samples from rounds greater than $t$, which are not yet assigned to any round or directly from round $t$. To see this, notice that the total number of samples in this iteration is $U_i = |\mathcal{F}|$. Therefore, for round $t$, we need at least $w_t |\mathcal{F}|$ samples. Those samples have to be from $F_t, F_{t+1}, \ldots, F_k$ and must not be assigned to another round $t' \neq t$. Assume that this already holds for all $t'' > t$. Then we only need to ensure that the amount of samples which are not yet assigned to any round plus the samples from round $t$ are greater equal $w_i |\mathcal{F}|$. This amount of not yet assigned samples plus the samples from round $t$ is equal to $|[F_{t+1}, \ldots, F_k]| - \sum_{t'=t+1}^{k} w_{t'} |\mathcal{F}| + |F_t|$. Item 2 follows from assuming that this is bigger than $w_t |\mathcal{F}|$.

We now prove Theorem 6 via a loop-invariant argument.

*Proof of Theorem 6.* Item 1 trivially holds, since only samples from $S_t$ are added to $F_t$.

We now show that item 2 also holds and that $|\mathcal{F}|$ is maximal. We define the following proposition $B_t$ for every $t \in \{1, \ldots, k\}$. $B_t$ holds iff for every $j \geq t$, it holds that

$$F_j \subseteq S_j \text{ and } |[F_j, \ldots, F_k]| \geq \sum_{t'=j}^{k} w_{t'} |\mathcal{F}| \tag{9}$$

We define the following loop invariant $C_t$. $C_t$ holds iff after iteration $t$ of the loop, $M'$ is the maximum integer such that $B_t$ holds and using $M' - |\mathcal{F}|$ samples for $F_1, \ldots, F_t$ does not lead to a violation of $B_t$.

If $C_t$ holds for every $t \in \{1, \ldots, k\}$, the theorem is shown.

---

**Algorithm 3:** Practical algorithm to compute the samples used by MDRR

---

1: **Input:** A sequence $S_1, \ldots, S_k$ of lists of samples and corresponding weights $w_1, \ldots, w_k \in \mathbb{R}^+$ such that $\sum_{t=1}^{k} w_t = 1$
2: **Output:** A sequence $\mathcal{F} = [F_1, \ldots, F_k]$ of lists of samples such that $F_t \subseteq S_t$, $|[F_t, \ldots, F_k]| \geq \sum_{t'=t}^{k} w_{t'} |\mathcal{F}|$ and $|\mathcal{F}|$ is maximal.

3: $M' \leftarrow +\infty$
4: Let $\mathcal{F}$ be a sequence of $k$ empty lists
5: **for** $t = k, \ldots, 1$ **do**
6:     **if** $M' - |\mathcal{F}| \leq |S_t|$ **then**
7:         Append $M' - |\mathcal{F}|$ samples from $S_t$ to $F_t$
8:         **Return** $\mathcal{F}$
9:     $F_t \leftarrow S_t$
10:     $W \leftarrow \sum_{t'=t}^{k} w_{t'}$
11:     $M' \leftarrow \left\lfloor \min\left( \frac{|\mathcal{F}|}{W}, M' \right) \right\rfloor$

---

We prove that $C_t$ holds via induction. $C_{k+1}$ holds before the loop starts, since we can think of $t$ to be equal to $k+1$ at this time, $M'$ is infinity and $\mathcal{F}$ empty.

The induction step goes from $t+1$ to $t$. Assume $C_{t+1}$ holds. The if-statement in line 6 then ensures that if there are more samples in $S_t$ than are still possible, $F_t$ is set equal to this number of samples and the algorithm returns. We know this is correct, since $M'$ is maximal. Otherwise $F_t$ is set to $S_t$, because this capacity is still there for samples from $S_t$.

Then in line 11, the $\min\left( \frac{|\mathcal{F}|}{W}, M' \right)$ defines the number of samples which can maximally be taken in total. The first argument of the minimum ensures that (9) holds for $j = t$. The second argument of the minimum, $M' + |\mathcal{F}|$ ensures that $B_{t+1}$ holds via the induction hypothesis. Then $B_t$ holds and the induction step is shown. $\qquad\square$

### B.3 SOLVING THE MIN-MAX OPTIMIZATION PROBLEM

In this subsection we describe how the learner solves the min-max problem (6) and the min-max problem in line 7 of Algorithm 2 in the experiments.

To solve the min-max problem of the empirical Lagrangians in equation (6) we use Algorithm 1 from Mandal et al. [2023].

To solve the min-max problem for MDRR (line 7 of Algorithm 2), we use Algorithm 4. It works the same as Algorithm 1 of Mandal et al. [2023], the only difference is in the conditions on $d$ in line 8, where we condition $d(s,a)/\overline{d}_t(s,a) \leq B$ for all steps since the last update of the policy. We use parameters, $N = 10$ and $\beta = \frac{\lambda}{2} = 0.05$.

---

**Algorithm 4:** FTRL algorithm to calculate an approximization for the finite sample optimization problem ((6) and (45))

---

1: **Input:** regularizing factor $\beta$, occupancy measures since the last update of the policy $\overline{d}_t$ for $t \in \{1, \ldots, k\}$
2: $d_0 \leftarrow \mathbf{0}$
3: **for** $j = 0, 1, \ldots, N-1$ **do**
4:     **if** $j = 0$ **then**
5:         $h_j \leftarrow \arg\min_h \hat{\mathcal{L}}^M(d_0, h) + \beta \|h\|_2^2$ s.t. $\|h\|_2 \leq \frac{3|S|}{(1-\gamma)^2}$
6:     **else**
7:         $h_j \leftarrow \arg\min_h \sum_{j'=1}^{j} \hat{\mathcal{L}}^M(d_{j'}, h) + \beta \|h\|_2^2$ s.t. $\|h\|_2 \leq \frac{3|S|}{(1-\gamma)^2}$
8:     $d_{j+1} \leftarrow \arg\max_d \hat{\mathcal{L}}^M(d, h_j)$ s.t. $\max_{s,a} d(s,a)/\overline{d}_t(s,a) \leq B$ for all $t \in \{1, \ldots, k\}$
9: **Return** $\sum_{j=1}^{N} d_j / N$

---

For the experiments with $w = 0.85$ and $w = 0.95$, we used machines with two AMD EPYC 7702 64-Core Processors and 2TB of RAM.

Table 2: Compute Times of the Experiments in Figures 1 and 3

| Algorithm | $k$ | $w$ | $v$ | time (rounded) |
|-----------|-----|-----|-----|----------------|
| RR | N\A | 0.5 | N\A | $\sim 94$ hrs |
| DRR | 3 | 0.5 | N\A | $\sim 78$ hrs |
| MDRR | 3 | 0.5 | 1.1 | $\sim 90$ hrs |
| RR | N\A | 0.15 | N\A | $\sim 105$ hrs |
| DRR | 3 | 0.15 | N\A | $\sim 88$ hrs |
| MDRR | 3 | 0.15 | 1.1 | $\sim 100$ hrs |
| MDRR | 5 | 0.5 | 1.1 | $\sim 85$ hrs |
| MDRR | 5 | 0.5 | 1.5 | $\sim 77$ hrs |
| MDRR | 5 | 0.5 | 1.8 | $\sim 78$ hrs |
| MDRR | 10 | 0.5 | 1.1 | $\sim 83$ hrs |

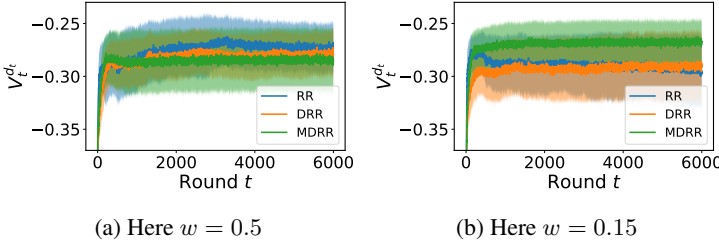

(a) Here $w = 0.5$      (b) Here $w = 0.15$

Figure 3: A sanity check if the algorithms reach valid solutions. Since the values of the three algorithms are close to one another, we assert that none of them reaches a much less optimal solution than another one, thereby validating all three approaches.

## B.4 TOTAL AMOUNT OF COMPUTE AND TYPE OF RESOURCES

The experiments of the main part (Figure 1) were run on a compute cluster with each machine having 4 Intel Xeon E7-8857 v2 CPUs (4 times 12 cores) and 1.5 TB of RAM.

In Table 2, we detail how long each experiment took to complete on these machines.

For the experiments with $w = 0.85$ and $w = 0.95$, we used machines with two AMD EPYC 7702 64-Core Processors and 2TB of RAM.

## B.5 SANITY-CHECK THE FAIRNESS OF THE COMPARISON

By only presenting Figures 1a and 1b in the main paper, we can not rule out that some of the algorithms converge to very suboptimal solutions. In this case the comparison would be unfair.

Therefore, in order to sanity-check the fairness of the comparison, we also investigate the expected value, $V_t^{d_t}$. This is not directly associated to finding a stable occupancy measure, but should rather be seen as a check to see if the algorithms we propose reach similar solutions. We compute $V_t^{d_t}$ using the rewards derived from the training sample trajectories. In other words, when $\mathrm{Tr}_t$ is the set of trajectories sampled in round $t$, and for each trajectory $\tau$, the reward in step $k$ is $r_t(\tau_k)$, then $V_t^{d_t} = \sum_{\tau \in \mathrm{Tr}_t} \sum_{k=0}^{l(\tau)} \gamma^k \cdot r_t(\tau_k)$. Here $l(\tau)$ is the length of trajectory $\tau$.

We see the expected values of the algorithms in Figure 3. As we see, after they settled down, the three algorithms have rather close expected values. We believe that the differences stem from the initialization of the environment of the second agent rather than from some inherent differences in the algorithms.

## B.6 ADDITIONAL RESULTS FOR LARGE VALUES OF $w$

We additionally ran experiments for larger values of $w$, in particular $w = 0.85$ and $w = 0.95$. The results are depicted in Figure 4. Suprisingly, we see that even with this large values of $w$, MDRR outperforms RR and DRR. This is somewhat counterintuitive, since at such large values of $w$ the environment is almost non-stateful, and we would expect RR to have an advantage here. We believe that this phenomenon is due to the fact that MDRR uses more samples than RR and DRR and therefore has a lower variance, even at the cost of a large bias. This seems to lead to a much better convergence in the settings we studied.

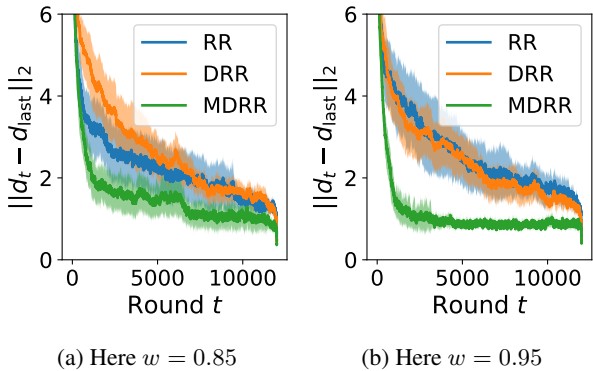

(a) Here $w = 0.85$        (b) Here $w = 0.95$

Figure 4: Convergence plots for less stationary environments, i.e. larger values of $w$. Data generated as in Figure 1. Also here MDRR outperforms the other algorithms.

## C ADDITIONAL THEORETICAL RESULTS

### C.1 EXAMPLE FOR ASSUMPTION 1

We now give an example to illustrate Assumption 1. For simplicity we assume that only the probability transition function $P$ changes and not the reward $r$. We consider a response model $\mathcal{P}$ which is defined in the following way:

$$\mathcal{P}(d, P, r) = wP + (1 - w)P^*(\pi_d)$$

for some decay rate $w \in (0, 1)$ and some response function $P^*(\pi_d)$. We can think of $\mathcal{P}$ being determined by a population and in each time-step a $(1 - w)$ fraction of the population responds to the newly deployed policy $\pi_d$. Similar settings have been studied in performative prediction [Ray et al., 2022].

We also assume that the change in $P^*$ is bounded, i.e. $|P^*(\pi_d)(s'|s, a) - P^*(\pi_{d'})(s'|s, a)| \leq c||d - d'||_2$ for all $s, s' \in S$, $a \in A$ and some constant $c > 0$. We can then derive the following Proposition.

**Proposition 1.** *With the conditions set in this subsection, it holds that*

$$||\mathcal{P}(d, P, r) - \mathcal{P}(d', P', r')||_2 \leq w||P - P'||_2 + (1 - w)c|S|\sqrt{|A|} \cdot ||d - d'||_2.$$

*Proof.*

$$\|\mathcal{P}(d, P, r) - \mathcal{P}(d', P', r')\|_2^2$$

$$= \sum_{s,a,s'} (w \cdot (P(s'|s,a) - P'(s'|s,a)) + (1 - w) \cdot P^*(\pi_d)(s'|a,s) - P^*(\pi_{d'})(s'|a,s))^2$$

$$\leq \sum_{s,a,s'} (w \cdot (P(s'|s,a) - P'(s'|s,a)) + (1 - w) \cdot c \cdot \|d - d'\|_2)^2$$

$$\leq \sum_{s,a,s'} (w \cdot (P(s'|s,a) - P'(s'|s,a)))^2 + \sum_{s,a,s'} ((1 - w) \cdot c \cdot \|d - d'\|_2)^2$$

$$+ \sum_{s,a,s'} 2((1 - w) \cdot c \cdot \|d - d'\|_2)(w \cdot (P(s'|s,a) - P'(s'|s,a)))$$

$$\leq w^2 \|P - P'\|_2^2 + |S|^2|A| \cdot ((1 - w)c)^2 \cdot \|d - d'\|_2^2 + 2 \cdot (1 - w) \cdot c \cdot \|d - d'\|_2 \cdot w \cdot \|P - P'\|_1$$

$$\leq w^2 \|P - P'\|_2^2 + |S|^2|A| \cdot ((1 - w)c)^2 \cdot \|d - d'\|_2^2 + 2|S|\sqrt{|A|} \cdot (1 - w) \cdot c \cdot \|d - d'\|_2 \cdot w \cdot \|P - P'\|_2$$

Taking the square root on both sides gives the desired result. □

Given this proposition, we choose $\epsilon_{p,p} = w$ and $\iota_p = (1 - w)c|S|\sqrt{|A|}$ (all other $\iota$ and $\epsilon$ parameters are 0). Now if $\iota_p = (1 - w)c|S|\sqrt{|A|} < 1$, Assumption 1 holds. That $(1 - w)c|S|\sqrt{|A|}$ is smaller than 1 is likely in many cases where $w$ is large and / or $c$ is small. The value of $w$ being large means that in each time-step, only a small fraction of the population responds to the new policy. This could likely be the case if each time-step encompasses a small amount of time. Additionally the total difference $\|P^*(\pi_d) - P^*(\pi_{d'})\|_1$ can be in the order of $c \cdot |S|^2|A| \cdot \|d - d'\|_2$, so the value of $c$ might likely be small.

## C.2 EXISTENCE OF STABLE POINTS

Using arguments similar to Mandal et al. [2023], we show that there exists a stable point.

**Proposition 2.** *If Assumption 1 holds, optimization problem* (2) *has a fixed point.*

*Proof.* This proposition is very similar to Proposition 1 from Mandal et al. [2023]. The proof follows theirs, and we don't repeat the arguments made in their proof. However in order to make use of their arguments, we need to show that $P_d$ and $r_d$ are continuous in $d$, which is not immediately clear. Recall that $P_d$ and $r_d$ map from occupancy measure $d$ to the environment the process converges to, if the learner always deploys $\pi_d$. We now prove that $P_d$ and $r_d$ are continuous in $d$.

We define $\epsilon := \max(\epsilon_p, \epsilon_r)$. Then we see that

$$\|P_d - P_{d'}\|_2 + \|r_d + r_{d'}\|_2 \leq \iota \|d - d'\|_2 + \epsilon (\|P_d - P_{d'}\|_2 + \|r_d - r_{d'}\|_2)$$

$$\leq \cdots \leq \sum_{i=0}^{\infty} \iota \epsilon^i \|d - d'\|_2 = \frac{\iota}{1 - \epsilon} \|d - d'\|_2 \tag{10}$$

The inequalities follow from Assumption 1. Thus $P_d$ and $r_d$ are continuous in $d$ and the rest of the proof follows from the same arguments as the proof of Proposition 1 from Mandal et al. [2023]. □

## C.3 APPROXIMATING THE UNREGULARIZED OBJECTIVE

Using arguments similar to Mandal et al. [2023], we can show the following approximation guarantee for the regularized objective.

**Theorem 7.** *For each setting RR, DRR and MDRR, when they approximate a stable policy $d_S$ with respect to the regularized objective* (3)*, the following guarantee holds:*

$$\sum_{s,a} r_{d_S}(s,a) \cdot d_S(s,a) \geq \max_{d \in \mathcal{C}(d_S)} \sum_{s,a} r_{d_S}(s,a) \cdot d(s,a) - \mathcal{O}\left(\frac{\lambda}{(1 - \gamma)^2}\right)$$

*Here $\mathcal{C}(d_S)$ denotes the set of occupancy measures which are feasible with respect to $P_{d_S}$.*

*Proof.* Since $d_S$ is a stable point with respect to objective (3), it holds that

$$\sum_{s,a} r_{d_S}(s,a) \cdot d_S(s,a) - \frac{\lambda}{2}\|d_S\|_2^2 \geq \max_{d \in \mathcal{C}(d_S)} \sum_{s,a} r_{d_S}(s,a) \cdot d(s,a) - \frac{\lambda}{2}\|d\|_2^2$$

Therefore,

$$\sum_{s,a} r_{d_S}(s,a) \cdot d_S(s,a) \geq \max_{d \in \mathcal{C}(d_S)} \sum_{s,a} r_{d_S}(s,a) \cdot d(s,a) - \frac{\lambda}{2}\|d\|_2^2$$

$$\geq \max_{d \in \mathcal{C}(d_S)} \sum_{s,a} r_{d_S}(s,a) \cdot d(s,a) - \frac{\lambda}{2(1-\gamma)^2}$$

The last inequality uses $\|d\|_2^2 = \sum_{s,a} d(s,a)^2 = (1-\gamma)^{-2} \sum_{s,a} ((1-\gamma)d(s,a))^2 \leq (1-\gamma)^{-2} \sum_{s,a}(1-\gamma)d(s,a) = (1-\gamma)^{-2}$. $\qquad\square$

## C.4 CONTRACTION

In contrast to the main paper, in the appendix $\epsilon$ refers to $\epsilon := \max(\epsilon_p, \epsilon_r)$, which signifies the dependency of the environment on the previous environment.

We define the following distances.

**Definition 3.** *For any occupancy measures $d, d'$, probability transition functions $P, P'$ and reward functions $r, r'$, we define the distance between $(d, P, r)$ and $(d', P', r')$ to be equal to*

$$\mathrm{dist}((d, P, r), (d', P', r')) := \|d - d'\|_2 + \|P - P'\|_2 + \|r - r'\|_2 \ .$$

*We overload notation to also define*

$$\mathrm{dist}((P, r), (P', r')) := \|P - P'\|_2 + \|r - r'\|_2 \ .$$

As described in section 2, show that the mapping from $(P, r)$ to the successor environment $(\mathcal{P}(d, P, r), \mathcal{R}(d, P, r))$ is a contraction.

**Proposition 3.** *Let $d$ be some occupancy measure. When Assumption 1 holds, in particular $\epsilon_p, \epsilon_r < 1$, the mapping $g_d(P, r) := (\mathcal{P}(d, P, r), \mathcal{R}(d, P, r))$ is a contraction with Lipschitz coefficient $\epsilon$.*

*Proof.* Let $P, P'$ be arbitrary probability transition functions and $r, r'$ arbitrary reward functions.

Then

$$\mathrm{dist}(g_d(P, r) - g_d(P', r')) = \|\mathcal{P}(d, P, r) - \mathcal{P}(d, P', r')\|_2 + \|\mathcal{R}(d, P, r) - \mathcal{R}(d, P', r')\|_2$$
$$\leq \epsilon_{p,p}\|P - P'\|_2 + \epsilon_{p,r}\|r - r'\|_2 + \epsilon_{r,p}\|P - P'\|_2 + \epsilon_{r,r}\|r - r'\|_2$$
$$\leq \epsilon\|P - P'\|_2 + \epsilon\|r - r'\|_2 = \epsilon \cdot \mathrm{dist}((P, r), (P', r')) \ .$$

Where the first inequality follows from Assumption 1 and the second one follows from the defintion of $\epsilon_p$, $\epsilon_r$ and $\epsilon$. From this the proposition follows. $\qquad\square$

## D PROOFS FOR REPEATED RETRAINING (RR) (SECTION 3)

### D.1 DEFINITIONS

We define the following numbers

**Definition 4.** *We define*

$$\alpha := \sqrt{3} + \frac{\sqrt{7}|S|\sqrt{|S|}}{(1-\gamma)^2} \text{ and}$$

$$\beta := \frac{(4\sqrt{7}\gamma + 3\sqrt{6})|S|}{(1-\gamma)^2} + \frac{18\sqrt{7}\gamma|S|^2\sqrt{|S|}}{(1-\gamma)^4}.$$

**Definition 5.** *Let* $\mathrm{GD}(P,r)$ *be the solution to the regularized optimization problem, with probability transition function* $P$ *and reward function* $r$*, i.e.*

$$\mathrm{GD}(P,r) := \arg\max_{d \geq 0} \sum_{s,a} d(s,a)r(s,a) - \frac{\lambda}{2}\|d\|_2^2$$

$$\text{s.t. } \sum_a d(s,a) = \rho(s) + \gamma \cdot \sum_{s',a} d(s',a)P(s',a,s) \,\forall s.$$

## D.2 RR IN THE EXACT SETTING (THEOREM 1)

We show the following more general version of Theorem 1.

**Theorem 8.** *Assume that Assumption 1 holds and*

$$\lambda > \max\left\{(1-\epsilon_p)^{-1}\beta, (1-\epsilon_r)^{-1}\alpha\right\}$$

*Then for any* $\delta > 0$*, we have*

$$\|d_t - d_S\|_2 \leq \delta \quad \text{for all } t \geq \frac{\ln\left(\frac{\|d_0-d_S\|_2+\|P_0-P_S\|_2+\|r_0-r_S\|_2}{\delta}\right)}{\ln\left(\left(\max\left\{\iota, \epsilon_p + \frac{\beta}{\lambda}, \epsilon_r + \frac{\alpha}{\lambda}\right\}\right)^{-1}\right)} + 1,$$

*with* $\alpha$ *and* $\beta$ *defined in Definition 4.*

We first discuss how to obtain Theorem 1 from Theorem 8. Assumption 2 ensures that $\beta \geq \alpha$, $\epsilon_p = \epsilon_r = \epsilon$ and $\iota \leq \epsilon$. We further bound $\|d_0 - d_S\|_2 \leq \frac{2}{1-\gamma}$, $\|P_0 - P_S\|_2 \leq \sqrt{2|S||A|}$ and $\|r_0 - r_S\|_2 \leq \sqrt{|S||A|}$. Choosing $\lambda = 2\beta(1-\epsilon)^{-1}$ then provides the desired bounds.

The proof of Theorem 8 has a similar structure to the proof of Theorem 4 in Brown et al. [2022].

*Proof of Theorem 8.* We define by $f$ the mapping from $(d_{t-1}, P_{t-1}, r_{t-1})$ to $(d_t, P_t, r_t)$, i.e.

$$f(d, P, r) := (\mathrm{GD}(P,r), \mathcal{P}(d, P, r), \mathcal{R}(d, P, r)).$$

We analyze $\mathrm{dist}(f(d, P, r), f(d', P', r'))$.

$$\mathrm{dist}(f(d, P, r), f(d', P', r')) = \|\mathrm{GD}(P,r) - \mathrm{GD}(P',r')\|_2$$
$$+ \|\mathcal{P}(d, P, r) - \mathcal{P}(d', P', r')\|_2 \tag{11}$$
$$+ \|\mathcal{R}(d, P, r) - \mathcal{R}(d', P', r')\|_2$$

The last two terms of this sum can be bounded by using Assumption 1 :

$$\|\mathcal{P}(d, P, r)) - \mathcal{P}(d', P', r'))\|_2 + \|\mathcal{R}(d, P, r)) - \mathcal{R}(d', P', r'))\|_2$$
$$\leq (\iota_p + \iota_r)\|d - d'\|_2 + (\epsilon_{p,p} + \epsilon_{r,p})\|P - P'\|_2 + (\epsilon_{p,r} + \epsilon_{r,r})\|r - r'\|_2 \tag{12}$$

We now bound the first term of (11), i.e. $\|\mathrm{GD}(P,r) - \mathrm{GD}(P',r')\|_2$.

From Lemma 1, we get

$$\| \operatorname{GD}(P, r) - \operatorname{GD}(P', r') \|_2 \leq \frac{\alpha}{\lambda} \| r - r' \|_2 + \frac{\beta}{\lambda} \| P - P' \|_2 \tag{13}$$

Combining (11), (12) and (13) we get

$$\begin{aligned}
\operatorname{dist}(f(d, P, r), f(d', P', r')) &\leq \iota_d \| d - d' \|_2 \\
&+ \left( \epsilon_p + \frac{\beta}{\lambda} \right) \| P - P' \|_2 + \left( \epsilon_r + \frac{\alpha}{\lambda} \right) \| r - r' \|_2
\end{aligned} \tag{14}$$

We define $q := \max \left( \iota_d, \epsilon_p + \frac{\beta}{\lambda}, \epsilon_r + \frac{\alpha}{\lambda} \right)$. From (14) and the definition of $q$, it follows that

$$\begin{aligned}
\operatorname{dist}((d_t, P_t, r_t), (d_S, P_S, r_S)) &= \operatorname{dist}(f(d_{t-1}, P_{t-1}, r_{t-1}), f(d_S, P_S, r_S)) \\
&\leq q \operatorname{dist}((d_{t-1}, P_{t-1}, r_{t-1}), (d_S, P_S, r_S)) \leq q^t \left( \| d_0 - d_S \|_2 + \| P_0 - P_S \|_2 + \| r_0 - r_S \|_2 \right),
\end{aligned}$$

where the first equality follows from the fact that $(d_S, P_S, r_S)$ is a fixed point of $f$.

Note that by the conditions on $\lambda, \iota, \epsilon_p$ and $\epsilon_r$, it holds that $q < 1$.

Therefore, if we set $t \geq \ln(\operatorname{dist}((d_1, P_0, r_0), (d_S, P_S, r_S))/\delta) / \ln(1/q) + 1$, then we get that

$$\operatorname{dist}((d_t, P_{t-1}, r_{t-1}), (d_S, P_S, r_S)) \leq \delta.$$

Then also $\| d_t - d_S \|_2 \leq \delta$. $\qquad \square$

**Lemma 1** (similar to lemma 2 of Brown et al. [2022]). *Let $P, \hat{P}$ be two probability transition functions and $r, \hat{r}$ be two reward functions. Then*

$$\| \operatorname{GD}(P, r) - \operatorname{GD}(\hat{P}, \hat{r}) \|_2 \leq \frac{\alpha}{\lambda} \| r - \hat{r} \|_2 + \frac{\beta}{\lambda} \| P - \hat{P} \|_2$$

*with $\alpha$ and $\beta$ from Definition 4.*

*Proof.* Let $M$ and $\hat{M}$ be two MDPs and $r$ and $\hat{r}$ be the corresponding reward functions and $P$ and $\hat{P}$ be the corresponding transition probability functions.

In the following we use some arguments from Mandal et al. [2023]. Those arguments apply here as well, since we use the same optimization problem as they do.

Let $h$ and $\hat{h}$ be the optimal solution to the dual objective (12) in Mandal et al. [2023] to $M$ and $\hat{M}$ respectively.

From Mandal et al. [2023] we get that (page 16, after "We now substitute the above bound in equation 15.")

$$-\frac{|A|(1-\gamma)^2}{\lambda} \left\| h - \hat{h} \right\|_2^2 \geq -\left\| h - \hat{h} \right\|_2 \left\| \nabla \mathcal{L}(\hat{h}; M) - \nabla \mathcal{L}(\hat{h}, \hat{M}) \right\|_2 \tag{15}$$

and also from Mandal et al. [2023]

$$\begin{aligned}
\left\| \nabla \mathcal{L}(\hat{h}; M) - \nabla \mathcal{L}(\hat{h}, \hat{M}) \right\|_2 &\leq \frac{4|S|\sqrt{|A|}}{\lambda} \| r - \hat{r} \|_2 + \left( \frac{4\gamma\sqrt{|S||A|}}{\lambda} + \frac{6\gamma\sqrt{|A|}|S|}{\lambda} \left\| \hat{h} \right\|_2 \right) \left\| P - \hat{P} \right\|_2 \\
&\leq \frac{4|S|\sqrt{|A|}}{\lambda} \| r - \hat{r} \|_2 + \left( \frac{4\gamma\sqrt{|S||A|}}{\lambda} + \frac{6\gamma\sqrt{|A|}|S|}{\lambda} \frac{3|S|}{(1-\gamma)^2} \right) \left\| P - \hat{P} \right\|_2
\end{aligned} \tag{16}$$

The first inequality is due to lemma 3 of Mandal et al. [2023] and the second inequality is due to lemma 4 in Mandal et al. [2023].

Combining (15) and (16) we get:

$$\left\|h - \hat{h}\right\|_2 \leq \frac{\lambda}{|A|(1-\gamma)^2} \left\|\nabla\mathcal{L}(\hat{h}; M) - \nabla\mathcal{L}(\hat{h}, \hat{M})\right\|_2$$

$$\leq \frac{\lambda}{|A|(1-\gamma)^2} \left( \frac{4|S|\sqrt{|A|}}{\lambda} \|r - \hat{r}\|_2 + \left( \frac{4\gamma\sqrt{|S||A|}}{\lambda} + \frac{6\gamma\sqrt{|A|}|S|}{\lambda} \frac{3|S|}{(1-\gamma)^2} \right) \left\|P - \hat{P}\right\|_2 \right) \quad (17)$$

Another result from Mandal et al. [2023], which is found in the proof of lemma 1 is:

$$\left\|\mathrm{GD}(P, r) - \mathrm{GD}(\hat{P}, \hat{r})\right\|_2^2 \leq \frac{3}{\lambda^2}\|r - \hat{r}\|_2^2 + \frac{7|A||S|}{\lambda^2}\left\|h - \hat{h}\right\|_2^2 + \frac{6}{\lambda^2}\left\|\hat{h}\right\|_2^2 \left\|P - \hat{P}\right\|_2^2 \quad (18)$$

Combining (17) and (18) it follows that

$$\left\|\mathrm{GD}(P, r) - \mathrm{GD}(\hat{P}, \hat{r})\right\|_2 \leq \frac{\sqrt{3}}{\lambda}\|r - \hat{r}\|_2 + \frac{\sqrt{7|A||S|}}{\lambda}\left\|h - \hat{h}\right\|_2 + \frac{\sqrt{6}}{\lambda}\left\|\hat{h}\right\|_2 \left\|P - \hat{P}\right\|_2$$

$$\leq \frac{\sqrt{3}}{\lambda}\|r - \hat{r}\|_2 + \frac{\sqrt{7|A||S|}}{\lambda}\left\|h - \hat{h}\right\|_2 + \frac{\sqrt{6}}{\lambda}\frac{3|S|}{(1-\gamma)^2}\left\|P - \hat{P}\right\|_2 \quad (19)$$

where the last inequality follows from lemma 4 of Mandal et al. [2023].

Combining (17) and (19) we get:

$$\left\|\mathrm{GD}(P, r) - \mathrm{GD}(\hat{P}, \hat{r})\right\|_2 \leq \frac{\sqrt{3}}{\lambda}\|r - \hat{r}\|_2 + \frac{\sqrt{7|A||S|}}{\lambda}\frac{\lambda}{|A|(1-\gamma)^2}\left( \frac{4|S|\sqrt{|A|}}{\lambda}\|r - \hat{r}\|_2 \right.$$

$$\left. + \left( \frac{4\gamma\sqrt{|S||A|}}{\lambda} + \frac{6\gamma\sqrt{|A|}|S|}{\lambda}\frac{3|S|}{(1-\gamma)^2} \right)\left\|P - \hat{P}\right\|_2 \right) + \frac{\sqrt{6}}{\lambda}\frac{3|S|}{(1-\gamma)^2}\left\|P - \hat{P}\right\|_2$$

$$= \left( \frac{\sqrt{3}}{\lambda} + \frac{\sqrt{7|A||S|}}{\lambda}\frac{\lambda}{|A|(1-\gamma)^2}\frac{4|S|\sqrt{|A|}}{\lambda} \right)\|r - \hat{r}\|_2$$

$$+ \left( \frac{\sqrt{7|A||S|}}{\lambda}\frac{\lambda}{|A|(1-\gamma)^2}\left( \frac{4\gamma\sqrt{|S||A|}}{\lambda} + \frac{6\gamma\sqrt{|A|}|S|}{\lambda}\frac{3|S|}{(1-\gamma)^2} \right) + \frac{\sqrt{6}}{\lambda}\frac{3|S|}{(1-\gamma)^2} \right)\left\|P - \hat{P}\right\|_2$$

$$= \left( \frac{\sqrt{3}}{\lambda} + \frac{\sqrt{7}|S|\sqrt{|S|}}{(1-\gamma)^2\lambda} \right)\|r - \hat{r}\|_2 + \left( \frac{(4\sqrt{7}\gamma + 3\sqrt{6})|S|}{(1-\gamma)^2\lambda} + \frac{18\sqrt{7}\gamma|S|^2\sqrt{|S|}}{(1-\gamma)^4\lambda} \right)\left\|P - \hat{P}\right\|_2$$

$\square$

## D.3 RR WITH FINITE SAMPLES (THEOREM 2)

In general we note that using our sample generation model, it is easy to get an estimate of the current occupancy measure $\bar{d}$, by comparing how many samples were drawn for each pair $(s, a)$ and how many samples were drawn overall. It is also straightforward to bound those estimates using standard methods such as Hoeffding's inequality. For simplicity, we implicitly assume that those occupancy measures are provided. More concretely, in Lagrangians (5) and (7) we assume that $\bar{d}_j$ is given.

**Definition 6.** *We denote by $\widehat{\mathrm{GD}}(d_t, F)$ the solution to optimization problem corresponding to $\hat{\mathcal{L}}$, i.e.*

$$\widehat{\mathrm{GD}}(d_t, F) := \arg\max_d \min_h \underbrace{\left( -\frac{\lambda}{2}\|d\|_2^2 + \sum_s h(s)\rho(s) + \sum_{(s,a,r,s')\in F} \frac{d(s,a)}{\bar{d}_t(s,a)} \cdot \frac{r - h(s) + \gamma h(s')}{|F|(1-\gamma)} \right)}_{=\hat{\mathcal{L}}}$$

We use the following result from Mandal et al. [2023].

**Lemma 2.** *Given an arbitrary occupancy measure $d$, probability transition function $P$ and reward function $r$, suppose that $\mathrm{GD}(P,r)(s,a)/\overline{d}(s,a) \leq B$ for all $(s,a) \in S \times A$, where $\overline{d}$ is the occupancy measure of $\pi_d$ in an environment with transition probabilities $P$. Furthermore, let $F$ be a set of samples drawn according to the occupancy measure $\overline{d}$ with $r$ being the reward function. We assume*

$$|F| \geq \frac{1}{\mu^2}\left(|A|\ln\left(\frac{2}{\delta_1}\right) + \ln\left(\frac{12|S|}{\mu(1-\gamma)^2}\right) + 2|A|\ln\left(\frac{\ln\left(\frac{3|S|^2|A|B}{\mu(1-\gamma)^2}\right)}{\mu}\right)\right),$$

*for arbitrary $\mu, \delta_1 > 0$. Then the following bound holds with probability at least $1 - \delta_1$.*

$$\|\mathrm{GD}(P,r) - \widehat{\mathrm{GD}}(d,F)\|_2 \leq \frac{6\sqrt{|S|^{1.5}(B + \sqrt{|A|})\mu}}{(1-\gamma)^{1.5}}\frac{1}{\sqrt{\lambda}}$$

This lemma follows from the equation which comes second after equation (23) in the work from Mandal et al. [2023] on page 30, after the text "Rearranging and using lemma 12 we get the following bound". The conditions follow from the conditions under which this equation holds in the work from Mandal et al. [2023]. Note that we write $\mu$ instead of $\epsilon$, which is the variable name used in Mandal et al. [2023]. The same arguments as in Mandal et al. [2023] hold, since they also look at the one step updates optimizing $\mathcal{L}$ and $\hat{\mathcal{L}}$, which are the same in this work.

We can then show a more general version of Theorem 2.

**Theorem 9.** *Suppose that overlap Assumption 3 holds for $k = 1$ and parameter $B$ and Assumption 1 holds. Let $(x_p, x_r) \in \{(\iota_p, \iota_r), (\epsilon_{p,p}, \epsilon_{r,p}), (\epsilon_{p,r}, \epsilon_{r,r})\}$ be the pair maximizing $\left(\frac{\alpha}{\lambda} + 1\right)x_r + \left(\frac{\beta}{\lambda} + 1\right)x_p$. We then assume that*

$$\lambda > \max\left\{(1-\epsilon_p)^{-1}\beta, (1-\epsilon_r)^{-1}\alpha, \frac{\alpha x_r + \beta x_p}{1 - \zeta - x_r - x_p}\right\}.$$

*Furthermore assume that*

$$m_t \geq \left(\frac{\xi}{\lambda\zeta^2}\right)^2\left(|A|\ln\left(\frac{4t^2}{p}\right) + \ln\left(\frac{12|S|\xi}{\lambda\zeta^2(1-\gamma)^2}\right) + 2|A|\ln\left(\frac{\xi\ln\left(\frac{3|S||A|B\xi}{\lambda\zeta^2(1-\gamma)^2}\right)}{\lambda\zeta^2}\right)\right),$$

*with $\xi = \frac{36|S|^{1.5}(B+\sqrt{|A|})}{\delta^2(1-\gamma)^3}$. Then for any $\delta > 0$, we have*

$$\|d_t - d_S\|_2 \leq \delta \quad \text{for all } t \geq \frac{\ln\left(\frac{\|d_1 - d_S\|_2 + \|P_0 - P_S\|_2 + \|r_0 - r_S\|_2}{\delta}\right)}{\ln\left(1/\left(\zeta + \left(\frac{\alpha}{\lambda} + 1\right)x_r + \left(\frac{\beta}{\lambda} + 1\right)x_p\right)\right)} + 1.$$

*Here $\zeta$ can be chosen to be an arbitrary value between $0$ and $1 - x_r - x_p$. It defines a trade-off between the conditions on the regularization parameter $\lambda$ and on the number of samples $m_t$.*

Theorem 2 follows from Theorem 9 in the following way. We set $\zeta = (1-\epsilon)/2$, and $\lambda = \frac{2\epsilon(\alpha+\beta)}{1-\epsilon}$.

Then for the denominator of the number of retrainings, we derive

$$\zeta + \left(\frac{\alpha}{\lambda} + 1\right)\frac{\epsilon}{2} + \left(\frac{\beta}{\lambda} + 1\right)\frac{\epsilon}{2} = \frac{1}{2} + \frac{\epsilon}{2} + \frac{\epsilon(\alpha+\beta)}{2\lambda} = \frac{1}{2} + \frac{\epsilon}{2} + \frac{1-\epsilon}{4} = \frac{3}{4} + \frac{\epsilon}{4}$$

We can bound $\|d_1 - d_S\|_2$, $\|P_0 - P_S\|_2$ and $\|r_0 - r_S\|_2$ like above, to obtain

$$t \geq \frac{\ln\left(\frac{\frac{2}{1-\gamma} + \left(1 + \sqrt{2}\right)\sqrt{|S||A|}}{\delta}\right)}{\ln\left(1/\left(\frac{3}{4} + \frac{\epsilon}{4}\right)\right)} + 1$$

*Proof of Theorem 9.* From lemma 2, we get that with probability $1 - \delta_1$

$$\| \operatorname{GD}(P_t, r_t) - \widehat{\operatorname{GD}}(d_t, F_t) \|_2 \leq \frac{6\sqrt{|S|^{1.5}(B + \sqrt{|A|})\mu}}{(1 - \gamma)^{1.5}} \frac{1}{\sqrt{\lambda}}, \tag{20}$$

as long as

$$m_t \geq \frac{1}{\mu^2} \left( |A| \ln \left( \frac{2}{\delta_1} \right) + \ln \left( \frac{12|S|}{\mu(1 - \gamma)^2} \right) + 2|A| \ln \left( \frac{\ln \left( \frac{3|S|^2|A|B}{\mu(1-\gamma)^2} \right)}{\mu} \right) \right).$$

If we set $\delta_1 = p/2t^2$ in step $t$, we get that event (20) holds with probability at least $1 - p/2t^2$ in round $t$. Via a union bound over all rounds, we get that event (20) holds with probability at least $1 - p$ in all rounds.

Let $\hat{g}(d_{t+1}, P_t, r_t)$ be the result after one round, i.e.

$$\hat{g}(d_{t+1}, P_t, r_t) = (\widehat{\operatorname{GD}}(d_{t+1}, F_{t+1}), \mathcal{P}(d_{t+1}, P_t, r_t), \mathcal{R}(d_{t+1}, P_t, r_t)) .$$

I.e. it holds that $(d_{t+2}, P_{t+1}, r_{t+1}) = \hat{g}(d_{t+1}, P_t, r_t)$.

We analyze

$$
\begin{aligned}
\operatorname{dist}(\hat{g}(d_{t+1}, P_t, r_t), (d_S, P_S, r_S)) = &\ \|\widehat{\operatorname{GD}}(d_{t+1}, F_{t+1}) - d_S\|_2 \\
&+ \|\mathcal{P}(d_{t+1}, P_t, r_t) - \mathcal{P}(d_S, P_S, r_S)\|_2 + \|\mathcal{R}(d_{t+1}, P_t, r_t) - \mathcal{R}(d_S, P_S, r_S)\|_2 \\
\leq &\ \|\widehat{\operatorname{GD}}(d_{t+1}, F_{t+1}) - d_S\|_2 \\
&+ \iota_d \|d_{t+1} - d_S\|_2 + \epsilon_p \|P_t - P_S\|_2 + \epsilon_r \|r_t - r_S\|_2
\end{aligned}
\tag{21}
$$

where the last inequality is due to Assumption 1.

It remains to analyze $\|\widehat{\operatorname{GD}}(d_{t+1}, F_{t+1}) - d_S\|_2$. Using equation (20), we see that

$$\|\widehat{\operatorname{GD}}(d_{t+1}, F_{t+1}) - d_S\|_2 \leq \|\widehat{\operatorname{GD}}(d_{t+1}, F_{t+1}) - \operatorname{GD}(P_{t+1}, r_{t+1})\|_2 + \|\operatorname{GD}(P_{t+1}, r_{t+1}) - d_S\|_2$$

$$\leq \frac{6\sqrt{|S|^{1.5}(B + \sqrt{|A|})\epsilon}}{(1 - \gamma)^{1.5}} \frac{1}{\sqrt{\lambda}} + \|\operatorname{GD}(P_{t+1}, r_{t+1}) - \operatorname{GD}(P_S, r_S)\|_2 \tag{22}$$

Furthermore we can derive

$$
\begin{aligned}
\|\operatorname{GD}(P_{t+1}, r_{t+1}) - \operatorname{GD}(P_S, r_S)\|_2 \leq &\ \frac{\alpha}{\lambda} \|r_{t+1} - r_S\|_2 + \frac{\beta}{\lambda} \|P_{t+1} - P_S\|_2 \\
= &\ \frac{\alpha}{\lambda} \|\mathcal{R}(d_{t+1}, P_t, r_t) - \mathcal{R}(d_S, P_S, r_S)\|_2 + \frac{\beta}{\lambda} \|\mathcal{P}(d_{t+1}, P_t, r_t) - \mathcal{P}(d_S, P_S, r_S)\|_2 \\
\leq &\ \frac{\alpha}{\lambda} (\iota_r \|d_{t+1} - d_S\|_2 + \epsilon_{r,p} \|P_t - P_S\|_2 + \epsilon_{r,r} \|r_t - r_S\|_2) \\
&+ \frac{\beta}{\lambda} (\iota_p \|d_{t+1} - d_S\|_2 + \epsilon_{p,p} \|P_t - P_S\|_2 + \epsilon_{p,r} \|r_t - r_S\|_2)
\end{aligned}
\tag{23}
$$

where the first inequality follows from lemma 1, in the equality we use the fact that $P_{t+1} = \mathcal{P}(d_{t+1}, P_t, r_t)$, $r_{t+1} = \mathcal{R}(d_{t+1}, P_t, r_t)$, $P_S = \mathcal{P}(d_S, P_S, r_S)$, $r_S = \mathcal{R}(d_S, P_S, r_S)$ and Assumption 1.

Inserting (23) into (22) and the result into (21), we get

$$
\begin{aligned}
\operatorname{dist}(\hat{g}(d_{t+1}, P_t, r_t), (d_S, P_S, r_S)) \leq &\ \frac{6\sqrt{|S|^{1.5}(B + \sqrt{|A|})\mu}}{(1 - \gamma)^{1.5}} \frac{1}{\sqrt{\lambda}} \\
&+ \left( \left( \frac{\alpha}{\lambda} + 1 \right) \iota_r + \left( \frac{\beta}{\lambda} + 1 \right) \iota_p \right) \|d_{t+1} - d_S\|_2 \\
&+ \left( \left( \frac{\alpha}{\lambda} + 1 \right) \epsilon_{r,p} + \left( \frac{\beta}{\lambda} + 1 \right) \epsilon_{p,p} \right) \|P_t - P_S\|_2 \\
&+ \left( \left( \frac{\alpha}{\lambda} + 1 \right) \epsilon_{r,r} + \left( \frac{\beta}{\lambda} + 1 \right) \epsilon_{p,r} \right) \|r_t - r_S\|_2
\end{aligned}
\tag{24}
$$

We now introduce a new parameter $\zeta \in (0, 1 - x_r - x_p)$, which is mentioned in the theorem. We set $\mu = \frac{\zeta^2 \delta^2 \lambda (1-\gamma)^3}{36|S|^{1.5}(B+\sqrt{|A|})}$.

This allows us to rewrite (24) into

$$
\begin{aligned}
\operatorname{dist}(\hat{g}(d_{t+1}, P_t, r_t), (d_S, P_S, r_S)) \leq & \zeta\delta \\
& + \left( \left( \frac{\alpha}{\lambda} + 1 \right) \iota_r + \left( \frac{\beta}{\lambda} + 1 \right) \iota_p \right) \|d_{t+1} - d_S\|_2 \\
& + \left( \left( \frac{\alpha}{\lambda} + 1 \right) \epsilon_{r,p} + \left( \frac{\beta}{\lambda} + 1 \right) \epsilon_{p,p} \right) \|P_t - P_S\|_2 \\
& + \left( \left( \frac{\alpha}{\lambda} + 1 \right) \epsilon_{r,r} + \left( \frac{\beta}{\lambda} + 1 \right) \epsilon_{p,r} \right) \|r_t - r_S\|_2 \\
\leq & \zeta\delta + \left( \left( \frac{\alpha}{\lambda} + 1 \right) x_r + \left( \frac{\beta}{\lambda} + 1 \right) x_p \right) \\
& \cdot (\|d_{t+1} - d_S\|_2 + \|P_t - P_S\|_2 + \|r_t - r_S\|_2)
\end{aligned}
\tag{25}
$$

Where we select $(x_p, x_r) \in \{(\iota_p, \iota_r), (\epsilon_{p,p}, \epsilon_{r,p}), (\epsilon_{p,r}, \epsilon_{r,r})\}$ to be the pair maximizing $\left( \frac{\alpha}{\lambda} + 1 \right) x_r + \left( \frac{\beta}{\lambda} + 1 \right) x_p$.

Note that by this formulation of $\mu$, the bound on $m_t$ becomes

$$
\begin{aligned}
m_t \geq & \left( \frac{36|S|^{1.5}(B + \sqrt{|A|})}{\zeta^2 \delta^2 \lambda (1-\gamma)^3} \right)^2 \left( |A| \ln \left( \frac{4t^2}{p} \right) + \ln \left( \frac{432|S|^{2.5}(B + \sqrt{|A|})}{\zeta^2 \delta^2 \lambda (1-\gamma)^5} \right) \right. \\
& + 2|A| \ln \left( \frac{36|S|^{1.5}(B + \sqrt{|A|}) \ln \left( \frac{108|S|^{3.5}|A|B(B+\sqrt{|A|})}{\zeta^2 \delta^2 \lambda (1-\gamma)^5} \right)}{\zeta^2 \delta^2 \lambda (1-\gamma)^3} \right) \Bigg)
\end{aligned}
$$

We now apply lemma 3 on the sequence $\{(d_{t+1}, P_t, r_t)\}_{t \in \mathbb{N}}$ using (25). We can do this, because by our assumption, we know that $\lambda > \frac{\alpha x_r + \beta x_p}{1 - \zeta - x_r - x_p}$ and $1 > \zeta + x_r + x_p$, so

$$
\begin{aligned}
& \zeta + \left( \frac{\alpha}{\lambda} + 1 \right) x_r + \left( \frac{\beta}{\lambda} + 1 \right) x_p \\
& < \zeta + x_r + x_p + \frac{\alpha x_r (1 - \zeta - x_r - x_p)}{\alpha x_r + \beta x_p} + \frac{\beta x_p (1 - \zeta - x_r - x_p)}{\alpha x_r + \beta x_p} = 1 \ .
\end{aligned}
$$

The bound stated in the Theorem follows by the application of lemma 3 and the fact that $\|d_{t+1} - d_S\|_2 \leq \operatorname{dist}((d_{t+1}, P_t, r_t), (d_S, P_S, r_S))$. $\qquad \square$

We use of the following argument, which is often used in the performative prediction setting [Perdomo et al., 2020, Brown et al., 2022, Mandal et al., 2023].

**Lemma 3.** *Let* $(\mathcal{M}, \operatorname{dist})$ *be a metric space and* $x_1, x_2 \geq 0$ *with* $x_1 + x_2 < 1$*. Assume that* $\{p_i\}_{i \in \mathbb{N}}$ *is a sequence of points in* $\mathcal{M}$ *such that there exists a unique* $p_S \in \mathcal{M}$ *with*

$$
\operatorname{dist}(p_{i+1}, p_S) \leq x_1 \delta + x_2 \operatorname{dist}(p_i, p_S) \quad \text{for all } i \geq 0 \ .
$$

*Then for* $n \geq \frac{\ln(\operatorname{dist}(p_0, p_S)/\delta)}{\ln(1/(x_1 + x_2))}$*, it holds that* $\operatorname{dist}(p_n, p_S) \leq \delta$.

*Proof.* We see this via the following case distinction. Let $i \geq 0$ be arbitrary.

Case 1: $\operatorname{dist}(p_i, p_S) \geq \delta$.
   Then

$$
\operatorname{dist}(p_{i+1}, p_S) \leq \operatorname{dist}(p_i, p_S)(x_1 + x_2) \ .
$$

Case 2: $\operatorname{dist}(p_i, p_S) < \delta$.
   Then

$$
\operatorname{dist}(p_{i+1}, p_S) \leq \delta(x_1 + x_2) \ .
$$

By this case distinction, via induction we get that $\text{dist}(p_i, p_S) \leq \max((x_1 + x_2)^i \text{dist}(p_0, p_S), \delta)$. In particular for $n = \frac{\ln(\text{dist}(p_0, p_S)/\delta)}{\ln(1/(x_1+x_2))}$ it holds that

$$\text{dist}(p_n, p_S) \leq \max((x_1 + x_2)^n \text{dist}(p_0, p_S), \delta) \leq \delta .$$

$\square$

# E  PROOFS FOR DELAYED REPEATED RETRAINING (DRR) (SECTION 4)

## E.1  DRR IN THE EXACT SETTING (THEOREM 3)

We show a more general version of the Theorem 3.

**Theorem 10.** *Suppose Assumption 1 holds and $\lambda > \frac{2\iota\phi}{1-\epsilon}$, where $\phi := \max(\alpha, \beta)$ and $\alpha, \beta$ as in Definition 4. Then with $d_i$ being calculated by DRR in the exact setting, with $k = \ln^{-1}\left(\frac{1}{\epsilon}\right) \ln\left(\frac{d_{P,r}}{\delta\iota}\right)$, it holds that*

$$\|d_i - d_S\|_2 \leq \delta \quad \text{for all } i \geq \ln\left(\frac{\|d_0 - d_S\|_2}{\delta}\right) / \ln\left(\frac{\lambda(1-\epsilon)}{2\phi\iota}\right) .$$

We first discuss how Theorem 3 follows from Theorem 10. Assumption 2 ensures that $\beta \geq \alpha$, $\epsilon_p = \epsilon_r = \epsilon$ and $\iota \leq \epsilon$. We bound $\|d_0 - d_S\|_2 \leq \frac{2}{1-\gamma}$. Choosing $\lambda = 2e\iota\beta(1-\epsilon)^{-1}$ then provides the desired bounds.

For proving Theorem 10, we use arguments similar to the ones Brown et al. [2022] use for proving Theorem 8.

*Proof of Theorem 10.* Let $P_0$ and $r_0$ be some arbitrary initial probability transition and reward function respectively. Denote by $(\tilde{P}_d, \tilde{r}_d)$ the transition probability and reward function after $k$ repeated deployments of $d$.

Note that $d_{i+1} = \text{GD}(\tilde{P}_{d_i}, \tilde{r}_{d_i})$ and $d_S = \text{GD}(P_S, r_S)$.

lemma 1 gives

$$\|d_{i+1} - d_S\|_2 = \| \text{GD}(\tilde{P}_{d_i}, \tilde{r}_{d_i}) - \text{GD}(P_S, r_S)\|_2 \leq \frac{\alpha}{\lambda}\|\tilde{r}_{d_i} - r_S\|_2 + \frac{\beta}{\lambda}\|\tilde{P}_{d_i} - P_S\|_2$$

$$\leq \frac{\phi}{\lambda}(\text{dist}((\tilde{P}_{d_i}, \tilde{r}_{d_i}), (P_S, r_S))) \tag{26}$$

We can decompose

$$\text{dist}((\tilde{P}_{d_i}, \tilde{r}_{d_i}), (P_S, r_S)) \leq \text{dist}((\tilde{P}_{d_i}, \tilde{r}_{d_i}), (P_{d_i}, r_{d_i})) + \text{dist}((P_{d_i}, r_{d_i}), (P_S, r_S)) \tag{27}$$

The first term of 27 can be bounded by lemma 5, the second term by lemma 4:

$$\text{dist}((\tilde{P}_{d_i}, \tilde{r}_{d_i}), (P_S, r_S)) \leq \frac{\iota}{1-\epsilon}\delta + \frac{\iota}{1-\epsilon}\|d_i - d_S\|_2 \tag{28}$$

Using 26 and 28 we get

$$\|d_{i+1} - d_S\|_2 \leq \frac{\phi\iota}{\lambda(1-\epsilon)}\delta + \frac{\phi\iota}{\lambda(1-\epsilon)}\|d_i - d_S\|_2 \tag{29}$$

We can apply lemma 3 on $\{d_i\}_{i\in\mathbb{N}}$, since $\frac{2\phi\iota}{\lambda(1-\epsilon)} < 1$ holds due to the assumptions on $\lambda$. Lemma 3, bounds the number of iterations until $d_i$ converges to a $\delta$ radius around $d_S$ and the statement of the theorem follows from this bound. $\square$

We now describe and prove the lemmas used in the proof of Theorem 10.

**Lemma 4** (similar to lemma 3 of Brown et al. [2022])**.** *Suppose Assumption 1 holds.*

*Let $d, d' \in D$ be arbitrary occupancy measures and let $P := P_d, r := r_d$ (and respectively $P' := P_{d'}, r' := r_{d'}$) be the probability transition and reward functions to which the system asymptotically converges, if $d$ (respectively $d'$) is applied repeatedly. It holds that*

$$\|P - P'\|_2 + \|r - r'\|_2 \leq \frac{\iota}{1 - \max(\epsilon_p, \epsilon_r)}\|d - d'\|_2 \tag{30}$$

*Proof.* Because of Assumption 1, it holds that

$$
\begin{aligned}
&\|P - P'\|_2 + \|r - r'\|_2 \\
&= \|\mathcal{P}(d, P, r) - \mathcal{P}(d', P', r')\|_2 + \|\mathcal{R}(d, P, r) - \mathcal{R}(d', P', r')\|_2 \\
&\leq \iota\|d - d'\|_2 + \epsilon_p\|P - P'\|_2 + \epsilon_r\|r - r'\|_2 \\
&\leq \iota\|d - d'\|_2 + \max(\epsilon_p, \epsilon_r)(\|P - P'\|_2 + \|r - r'\|_2)
\end{aligned}
$$

Where the equality holds because $(P, r)$ and $(P', r')$ are the long-term transition probabilities and reward functions for $d$ and $d'$ respectively. The inequality holds because of Assumption 1.

The statement of the lemma follows from this equation. $\qquad\square$

**Lemma 5** (similar to lemma 4 of Brown et al. [2022])*. Assume Assumption 1 holds with $\epsilon_p, \epsilon_r < 1$. Given a policy $\pi$, denote by $(\tilde{P}_\pi, \tilde{r}_\pi)$ the transition probability and reward function after $k = \ln^{-1}(\frac{1}{\epsilon}) \ln\left(\frac{\text{dist}((P_0, r_0), (P_1, r_1))}{\nu}\right)$ deployments of $\pi$, for any initial probability transition function $P_0$ and reward function $r_0$. It holds that*

$$
\left\|P_\pi - \tilde{P}_\pi\right\|_2 + \|r_\pi - \tilde{r}_\pi\|_2 \leq \frac{\nu}{1 - \epsilon} \; .
$$

*Proof.* Using Proposition 3, we see that

$$
\begin{aligned}
&\text{dist}((\tilde{P}_\pi, \tilde{r}_\pi), (P_\pi, r_\pi))(1 - \epsilon) \leq \epsilon^k \, \text{dist}((P_0, r_0), (P_\pi, r_\pi))(1 - \epsilon) \\
&\leq \epsilon^k \, \text{dist}((P_0, r_0), (P_\pi, r_\pi)) - \epsilon^k \, \text{dist}((P_1, r_1), (P_\pi, r_\pi)) \\
&\leq \epsilon^k \, \text{dist}((P_0, r_0), (P_1, r_1)).
\end{aligned}
$$

Therefore

$$
\text{dist}((\tilde{P}_\pi, \tilde{r}_\pi), (P_\pi, r_\pi)) \leq \frac{\epsilon^k}{1 - \epsilon} \, \text{dist}((P_0, r_0), (P_1, r_1)). \tag{31}
$$

Using $k \geq \ln^{-1}\left(\frac{1}{\epsilon}\right) \ln\left(\frac{\text{dist}((P_0, r_0), (P_1, r_1))}{\nu}\right)$, we get $\epsilon^k \leq \frac{\nu}{\text{dist}((P_0, r_0), (P_1, r_1))}$. If we insert this into 31, we get the desired bound. $\qquad\square$

## E.2 DRR WITH FINITE SAMPLES (THEOREM 4)

We show a more general version of the Theorem 4.

**Theorem 11.** *Let $d_i$ be computed by finite sample DRR with $k = \ln^{-1}\left(\frac{1}{\epsilon}\right) \ln\left(\frac{5\,\mathrm{d}_{P,r}}{\delta\iota}\right)$. Suppose Assumption 1 holds and Assumption 3 holds for $k$ and parameter $B$. Furthermore assume $\lambda > \max\left(5.76\xi\mu, \xi\mu + \frac{\iota\phi}{(1-\epsilon)}\left(1 + \frac{1}{4.8\xi\mu}\right)\right)$, with $\xi$ as defined above. Furthermore assume that*

$$
\begin{aligned}
m_i \geq \frac{1}{\mu^2} \Bigg( &|A| \ln\left(\frac{4i^2}{p}\right) + \ln\left(\frac{12|S|}{(1-\gamma)^2\mu}\right) \\
&+ 2|A| \ln\left(\frac{\ln\left(\frac{3|S|^2|A|B}{(1-\gamma^2)\mu}\right)}{\mu}\right) \Bigg) .
\end{aligned}
$$

*Then for any $\delta > 0$, we have*

$$
\|d_i - d_S\|_2 \leq \delta
$$

$$
\text{for all } i \geq \frac{\ln\left(\frac{\|d_1 - d_S\|_2}{\delta}\right)}{\ln\left(\left(\sqrt{\frac{\xi\mu}{\lambda}} + \frac{1.2\iota\phi}{\lambda(1-\epsilon)}\right)^{-1}\right)} + 1.
$$

*Here $\mu > 0$ can be chosen arbitrarily and defines a trade-off between the conditions on the number of samples $m_i$ and on the regularization factor $\lambda$.*

We first show how Theorem 4 follows from Theorem 11. Assumption 2 ensures that $\beta \geq \alpha$, $\epsilon_p = \epsilon_r = \epsilon$, $\iota \leq \epsilon$ and $\beta \geq \alpha$
For Theorem 4, we use $\mu = \frac{\lambda}{10\xi}$. We bound

$$\lambda > \max\left(1, \frac{40\iota\beta}{9(1-\epsilon)}, \frac{1.2\iota\beta}{(1-\epsilon)\left(\frac{\epsilon}{4} + \frac{3}{4} - \frac{1}{\sqrt{10}}\right)}\right)$$

We then derive the bound on $i$ in the following way. For the denominator of the bound on $i$, we can then derive $\sqrt{\frac{\xi\mu}{\lambda}} + \frac{1.2\iota\phi}{\lambda(1-\epsilon)} = \frac{1}{\sqrt{10}} + \frac{1.2\iota\beta}{\lambda(1-\epsilon)} \leq \frac{\epsilon+3}{4}$. For the numerator of the bound on $i$, we use $\|d_1 - d_S\|_2 \leq \frac{2}{1-\gamma}$.

The bound on the number of samples follows from the fact that $\frac{1}{\mu^2} = \frac{100\xi^2}{\lambda^2} = \mathcal{O}\left(\frac{|S|^3(B+\sqrt{|A|})^2}{\delta^4(1-\gamma)^6\lambda^2}\right)$.

*Proof of Theorem 11.* In general, we bound

$$\|d_{i+1} - d_S\|_2 \leq \underbrace{\|d_{i+1} - d^*_{i+1}\|_2}_{T_1} + \underbrace{\|d^*_{i+1} - d_S\|_2}_{T_2} \tag{32}$$

where $d^*_{i+1}$ is the occupancy measure optimizing the exact Lagrangian after $k$ deployments of $\pi_{d_i}$, i.e. $d^*_{i+1} = \mathrm{GD}(P_{(i+1)\cdot k}, r_{(i+1)\cdot k})$.

We can apply lemma 2, since Assumption 3 holds. Let $F_t$ be the samples of round $t$. By setting $\delta_1 = p/2i^2$ we get with probablility at least $1 - p/2i^2$ in step $i$,

$$T_1 = \left\|\widehat{\mathrm{GD}}(d_i, F_{(i+1)\cdot k}) - \mathrm{GD}(P_{(i+1)\cdot k}, r_{(i+1)\cdot k})\right\|_2 \leq \frac{6\sqrt{|S|^{1.5}(B+\sqrt{|A|})\mu}}{(1-\gamma)^{1.5}}\frac{1}{\sqrt{\lambda}}, \tag{33}$$

if

$$\left|F_{(i+1)\cdot k}\right| \geq \frac{1}{\mu^2}\left(|A|\ln(4i^2/p) + \ln(12|S|/((1-\gamma)^2\mu)) + 2|A|\ln(\ln(3|S|^2|A|B/((1-\gamma^2)\mu))/\mu)\right)$$

By a union bound over all rounds, we get that (33) holds with probability $1 - p$ for every $i \in \mathbb{N}$.

To bound $T_2$, we can apply lemma 7 with parameter $\nu$, to get

$$T_2 = \left\|d^*_{i+1} - d_S\right\|_2 \leq \frac{\phi\nu}{\lambda(1-\epsilon)} + \frac{\phi\iota}{\lambda(1-\epsilon)}\|d_i - d_S\|_2 .$$

We determine $\nu$ later in the proof.

Inserting those bounds on $T_1$ and $T_2$ into (32), we get

$$\|d_{i+1} - d_S\|_2 \leq \frac{6\sqrt{|S|^{1.5}(B+\sqrt{|A|})\mu}}{(1-\gamma)^{1.5}}\frac{1}{\sqrt{\lambda}} + \frac{\phi\nu}{\lambda(1-\epsilon)} + \frac{\phi\iota}{\lambda(1-\epsilon)}\|d_i - d_S\|_2 = x_1\delta + x_2\|d_i - d_S\|_2 \tag{34}$$

where we define $x_1 := \frac{6\sqrt{|S|^{1.5}(B+\sqrt{|A|})\mu}}{(1-\gamma)^{1.5}\sqrt{\lambda}\delta} + \frac{\phi\nu}{\lambda(1-\epsilon)\delta}$ and $x_2 := \frac{\phi\iota}{\lambda(1-\epsilon)}$.

Note that we can write $x_1 + x_2$ as follows

$$x_1 + x_2 = \frac{6\sqrt{|S|^{1.5}(B+\sqrt{|A|})\mu}}{(1-\gamma)^{1.5}\sqrt{\lambda}\delta} + \frac{(\frac{\nu}{\delta}+\iota)\phi}{\lambda(1-\epsilon)} \tag{35}$$

We now derive conditions on $\lambda$ for when $x_1 + x_2 < 1$, because then we can apply lemma 3 to bound the iterations until which the sequence of $\{d_i\}_{i\in\mathbb{N}_{\geq 1}}$ converges. To this end, we can apply lemma 6 with $x = \lambda$, $a = \frac{6\sqrt{|S|^{1.5}(B+\sqrt{|A|})\mu}}{(1-\gamma)^{1.5}\delta}$, $b = \frac{(\frac{\nu}{\delta}+\iota)\phi}{1-\epsilon}$ and $y = 2.4$, to get that $x_1 + x_2 < 1$ holds, if

$$\lambda > \max\left(5.76a^2, a^2 + \frac{(\frac{\nu}{\delta}+\iota)\phi}{1.2(1-\epsilon)} + \frac{(\frac{\nu}{\delta}+\iota)\phi}{2.4^2(1-\epsilon)a^2}\right) .$$

We get the bound for $\lambda$ stated in the Theorem by setting $\nu = 0.2\delta\iota$.

Thus we can apply lemma 3 on the sequence $\{d_i\}_{i \in \mathbb{N}_{\geq 1}}$, to see that if $i \geq \ln\left(\frac{\|d_1 - d_S\|_2}{\delta}\right) / \ln\left(1/(x_1 + x_2)\right) + 1$, it holds that $\|d_i - d_S\|_2 \leq \delta$. The Theorem then follows from substituting $x_1 + x_2$ using equation (35) and $\nu = 0.2\delta\iota$. $\qquad\square$

For the proof, we used the following lemmas.

**Lemma 6.** *Let $a, b, x \geq 0$ and $y > 0$ be arbitrary. If $x > \max(y^2 a^2, a^2 + \frac{2b}{y} + \frac{b}{y^2 a^2})$, it holds that*

$$1 > \frac{a}{\sqrt{x}} + \frac{b}{x} . \tag{36}$$

*Proof.* When we multiply both sides of (36) with $\sqrt{x}$ and square the resulting term, we see that (36) is equivalent to

$$x > a^2 + 2\frac{ab}{\sqrt{x}} + \frac{b}{x} . \tag{37}$$

If we assume that $x > y^2 a^2$, we get

$$a^2 + 2\frac{ab}{\sqrt{x}} + \frac{b}{x} < a^2 + \frac{2b}{y} + \frac{b}{y^2 a^2} < x,$$

This shows that equation 37 and thus also equation 36 hold. $\qquad\square$

**Lemma 7.** *Suppose Assumption 1 holds with $\iota_d, \epsilon_p, \epsilon_r < 1$. Let $d$ be some occupancy measure and $P_0, r_0$ be some initial probability transition and reward functions. Let $P_t, r_t$ be the probability transition and reward function after $t > 0$ deployments of $\pi_d$. Then for $d' = \mathrm{GD}(P_k, r_k)$ with $k = \ln^{-1}\left(\frac{1}{\epsilon}\right) \ln\left(\frac{\mathrm{dist}((P_0, r_0), (P_1, r_1))}{\nu}\right)$, it holds that*

$$\|d' - d_S\| \leq \frac{\phi\nu}{\lambda(1 - \epsilon)} + \frac{\phi\iota}{\lambda(1 - \epsilon)} \|d - d_S\|_2 ,$$

*where $\phi = \max(\alpha, \beta)$ and $\alpha, \beta$ from Definition 4.*

*Proof.* Note that

$$\|d' - d_S\|_2 \leq \|d' - \mathrm{GD}(P_d, r_d)\|_2 + \|\mathrm{GD}(P_d, r_d) - d_S\|_2 \tag{38}$$

Using lemmas 1 and 5, we can bound

$$\|d' - \mathrm{GD}(P_d, r_d)\|_2 \leq \frac{\phi}{\lambda} \mathrm{dist}((P_k, r_k), (P_d, r_d)) \leq \frac{\phi\nu}{\lambda(1 - \epsilon)} . \tag{39}$$

Furthermore, by lemmas 1 and 4 we see that

$$\|\mathrm{GD}(P_d, r_d) - d_S\|_2 \leq \frac{\phi}{\lambda} \mathrm{dist}((P_d, r_d), (P_S, r_S)) \leq \frac{\phi\iota}{\lambda(1 - \epsilon)} \|d - d_S\|_2 . \tag{40}$$

Inserting (39) and (40) into (38) gives the desired bound. $\qquad\square$

# F PROOF FOR MDRR (THEOREM 5)

## F.1 PREPARATIONS FOR THE PROOF

For our derivations, we need an exact version of the empirical Lagrangian (7). To this end, consider the following optimization problem, which works with multiple reward and probability transition functions from different rounds.

$$\max_{d \geq 0} \sum_{s,a} d(s,a) \overline{r}_i(s,a) - \frac{\lambda}{2} \|d\|_2^2 \tag{41}$$

$$\text{s.t.} \sum_a d(s,a) = \rho(s) + \gamma \cdot \sum_{s',a} d(s',a) \overline{P}_i(s',a,s) \ \forall s$$

where we define $\overline{r}_i := \sum_{g=1}^k \frac{m_{ik+g}}{U_i} r_{ik+g}$ and $\overline{P}_i := \sum_{g=1}^k \frac{m_{ik+g}}{U_i} P_{ik+g}$ where $m_{ik+g} \geq 0$ is arbitrary and $U_i = \sum_{g=1}^k m_{ik+g}$. Equation (41) defines an objective for a mixture of probability transition and reward functions of the rounds in which the learner repeatedly deployed $\pi_{d_i}$. Each reward and probability transition is weighted by a weight $\frac{m_{ik+g}}{U_i}$. This optimization problem does not use finite samples, but the true reward and probability transition functions.

We can now show that the Lagrangian of (41) looks similar to the empirical Lagrangian (7) of MDRR.

$$\mathcal{L}^M(d,h,i) = d^\top \overline{r}_i - \frac{\lambda}{2} \|d\|_2^2 + \sum_s h(s) \left( -\sum_a d(s,a) + \rho(s) + \gamma \cdot \sum_{s',a} d(s',a) \overline{P}_i(s|s',a) \right)$$

$$= d^\top \sum_{g=1}^k \frac{m_{ik+g}}{U_i} r_i^g - \frac{\lambda}{2} \|d\|_2^2 + \sum_s h(s) \left( -\sum_a d(s,a) + \rho(s) + \gamma \cdot \sum_{s',a} d(s',a) \sum_{g=1}^k \frac{m_{ik+g}}{U_i} P_{i \cdot k+g}(s|s',a) \right)$$

$$= -\frac{\lambda}{2} \|d\|_2^2 + \sum_s h(s)\rho(s) + \sum_{g=1}^k \sum_{s,a} \frac{m_{ik+g}}{U_i} d(s,a) \left( r_{ik+g}(s,a) - h(s) + \gamma \sum_{s'} P_{i \cdot k+g}(s'|s,a)h(s') \right)$$

$$= -\frac{\lambda}{2} \|d\|_2^2 + \sum_s h(s)\rho(s) + \sum_{g=1}^k \sum_{s,a} \overline{d}_{ik+g}(s,a) \frac{m_{ik+g}}{U_i} \frac{d(s,a)}{\overline{d}_{ik+g}(s,a)} \left( r_{ik+g}(s,a) - h(s) + \gamma \sum_{s'} P_{i \cdot k+g}(s'|s,a)h(s') \right) \tag{42}$$

We then show a kind of closeness of $\mathcal{L}^M$ and $\hat{\mathcal{L}}^M$ in the following lemma. The lemma is a more general version of lemma 10 from Mandal et al. [2023]. The proof ideas follow theirs.

**Lemma 8.** *Suppose we are given an occupancy measure $d$ with $\max_{s,a} d(s,a)/\overline{d}_{ik+g}(s,a) \leq B$ for all $g \in [k]$, an $\|h\|_2 \leq H$ and $m_{ik+g} = w_g U_i$ with $U_i \geq \frac{1}{\eta^2} \left( |A| \ln \left( \frac{2\ln(|S||A|BH/\eta)}{\eta} \right) + \ln \left( 1 + \frac{2H}{\eta} \right) + \frac{\ln(2/\delta_1)}{|S|} \right)$. Furthermore assume $w_g \geq 0$ and $\sum_{g=1}^k w_g = 1$. Then the following bound holds with probability at least $1 - \delta_1$.*

$$\left| \hat{\mathcal{L}}^M(d,h;i) - \mathcal{L}^M(d,h;i) \right| \leq \frac{6(H+1)\sqrt{|S|}(B + \sqrt{|A|})\eta}{1 - \gamma} \ .$$

*for any $\eta > 0$.*

*Proof.* For this proof to simplify notation, we drop the '$i \cdot k$' in the subscript, and only use $g$, since we always consider the same iteration $i$.

Note that $\frac{m^g}{M} = w_g$.

We see that the expected value of the $\hat{\mathcal{L}}^M$ equals $\mathcal{L}^M$ as follows

$$\mathbb{E}[\hat{\mathcal{L}}^M(d,h,i)]$$

$$= -\frac{\lambda}{2}\|d\|_2^2 + \sum_s h(s)\rho(s) + \sum_{g=1}^{k}\sum_{l=1}^{|F_g|}\frac{1}{|F_g|}w_g\mathbb{E}_{(s,a,s')\sim M^g}\left[\frac{d(s,a)}{\overline{d}_g(s,a)}\cdot\frac{r_g(s,a)-h(s)+\gamma h(s')}{1-\gamma}\right]$$

$$= -\frac{\lambda}{2}\|d\|_2^2 + \sum_s h(s)\rho(s) + \sum_{g=1}^{k}\sum_{s,a}\overline{d}_g(s,a)w_g\frac{d(s,a)}{\overline{d}_g(s,a)}\left(r_g(s,a)-h(s)+\gamma\sum_{s'}P_g(s'|s,a)h(s')\right)$$

$$=\mathcal{L}^M(d,h,i)$$

where we use the notation $(s,a,s')\sim M^g$ to indicate that the tuple $(s,a,s')$ is distributed via the MDP in round $g$ of this iteration.

By the assumptions of this lemma, we see that

$$\left|\frac{1}{1-\gamma}\frac{d(s,a)}{\overline{d}_g(s,a)}(r_g(s,a)-h(s)+\gamma h(s'))\right| \leq \frac{B(H(1+\gamma)+1)}{1-\gamma}$$

By this, we can apply Hoeffding's inequality to get

$$\mathbb{P}\left(\left|\hat{\mathcal{L}}(d,h,i)-\mathcal{L}(d,h,i)\right| \geq \frac{2B(H(1+\gamma)+1)}{1-\gamma}\sqrt{\sum_{g=1}^{k}\frac{w_g^2}{m_t}\frac{\ln(2/\delta_1)}{2}}\right) \leq \delta_1$$

We now extend this bound to any occupancy measure $d$ and $h\in\mathcal{H}=\{h:\|h\|_2\leq H\}$. In order to do this, we first construct an $\eta$-net for the set of possible $h$s, $\mathcal{H}:=\{h\in\mathbb{R}^{|S|}:\|h\|_2\leq H\}$ and for the set of possible occupancy measures $\mathcal{D}$ which formally equals $\mathcal{D}=\left\{d:\frac{d(s,a)}{\overline{d}_g(s,a)}\leq B \text{ for all }(s,a,g)\in|S|\times|A|\times[k]\right\}$.

For $\mathcal{H}$, we can use lemma 5.2 from Vershynin [2010] to get a set $\mathcal{H}_\eta$ of size at most $\left(1+\frac{2H}{\eta}\right)^{|S|}$, such that for all $h\in\mathcal{H}$, there exists an $h_\eta\in\mathcal{H}_\eta$ for which it holds that $\|h-h_\eta\|_2\leq\eta$.

For $\mathcal{D}$ we choose a multiplicative $\eta$-net as follows. For each pair $(s,a)$ we choose grid points $\overline{d}_g(s,a)$, $(1+\eta)\overline{d}_g(s,a)$, $\ldots,(1+\eta)^p\overline{d}_g(s,a)$ with $p=\frac{\ln(B/\overline{d}_g(s,a))}{\ln(1+\eta)}$. Note that $\overline{d}_g$ could be arbitrarily small, but without loss of generality, we can assume that $\overline{d}_g(s,a)\geq\frac{\eta}{4|S||A|BH}$. This is because if we ignore all $(s,a,g)$ tuples in the sum in the second line of term (42), the error we introduce to $\mathcal{L}^M$ is at most $\eta/4$. Using this insight, we can thus choose $p=\frac{2\ln(|S||A|BH/\eta)}{\ln(1+\eta)}$. So we can choose an $\eta$-net $\mathcal{D}_\eta$ of size at most $\left(\frac{2\ln(|S||A|BH/\eta)}{\ln(1+\eta)}\right)^{|S||A|}\leq\left(\frac{2\ln(|S||A|BH/\eta)}{\eta}\right)^{|S||A|}$, such that for every $d\in\mathcal{D}$, there exists an $\tilde{d}\in\mathcal{D}_\eta$ such that $\frac{d(s,a)}{\tilde{d}(s,a)}\leq B$.

With a union bound over the elements of $\mathcal{H}_\eta$ and $\mathcal{D}_\eta$, we have that for all $d\in\mathcal{D}_\eta$ and $h\in\mathcal{H}_\eta$,

$$\mathbb{P}\Bigg(\left|\hat{\mathcal{L}}^M(d,h,i)-\mathcal{L}^M(d,h,i)\right|$$

$$\geq\frac{B(H(1+\gamma)+1)}{1-\gamma}\sqrt{\sum_{g=1}^{k}\frac{w_g^2}{m_g}\left(|S||A|\ln\left(\frac{2\ln(|S||A|BH/\eta)}{\eta}\right)+|S|\ln\left(1+\frac{2H}{\eta}\right)+\ln\left(\frac{2}{\delta_1}\right)\right)}\Bigg)\leq\delta_1 \tag{43}$$

We next extend this bound to all elements in $\mathcal{D}$ and $\mathcal{H}$. For every $d\in\mathcal{D}$ and $h\in\mathcal{H}$ there exits $\tilde{d}\in\mathcal{D}_\eta$ and $\tilde{h}\in\mathcal{H}_\eta$ such that $\max_{s,a}d(s,a)/\tilde{d}(s,a)\leq\eta$ and $\left\|h-\tilde{h}\right\|_2\leq\eta$. Let $\mathcal{L}_0^M(d,h;i)=\mathcal{L}^M(d,h;i)+\frac{\lambda}{2}\|d\|_2^2-\sum_s h(s)\rho(s)$ and $\hat{\mathcal{L}}_0^M(d,h;i)$ analogously.

Then

$$\left|\hat{\mathcal{L}}^M(d,h;i)-\mathcal{L}^M(d,h;i)\right| \leq \left|\hat{\mathcal{L}}_0^M(d,h;i)-\hat{\mathcal{L}}_0^M(\tilde{d},\tilde{h};i)\right|$$

$$+\left|\hat{\mathcal{L}}^M(\tilde{d},\tilde{h};i)-\mathcal{L}^M(\tilde{d},\tilde{h};i)\right|+\left|\mathcal{L}_0^M(\tilde{d},\tilde{h};i)-\mathcal{L}_0^M(d,h;i)\right| \tag{44}$$

Using lemma 11 from Mandal et al. [2023] we can bound

$$\left|\hat{\mathcal{L}}_0^M(d,h;i) - \hat{\mathcal{L}}_0^M(\tilde{d},\tilde{h};i)\right| = \sum_{g=1}^k w_g \left|\left( \sum_{(s,a,r,s')\in F_g} \frac{d(s,a)}{\overline{d}_g(s,a)} \frac{r - h(s) + \gamma \sum_{s'} h(s')}{m^g(1-\gamma)} \right.\right.$$
$$\left.\left. - \sum_{(s,a,r,s')\in F_g} \frac{\tilde{d}(s,a)}{\overline{d}_g(s,a)} \frac{r - \tilde{h}(s) + \gamma \sum_{s'} \tilde{h}(s')}{m^g(1-\gamma)} \right)\right|$$
$$\leq \frac{4BH\sqrt{|S|}\eta}{1-\gamma}$$

and

$$\left|\mathcal{L}_0^M(d,h;i) - \mathcal{L}_0^M(\tilde{d},\tilde{h};i)\right| = \left| \sum_{s,a} d(s,a) \left( \underbrace{\sum_{g=1}^k w_g r_g(s,a)}_{=\overline{r}(s,a)} - h(s) + \gamma \sum_{s'} h(s') \underbrace{\sum_{g=1}^k w_g P_g(s'|s,a)}_{=\overline{P}(s'|s,a)} \right) \right.$$
$$\left. - \sum_{s,a} \tilde{d}(s,a) \left( \underbrace{\sum_{g=1}^k w_g r_g(s,a)}_{=\overline{r}(s,a)} - \tilde{h}(s) + \gamma \sum_{s'} \tilde{h}(s') \underbrace{\sum_{g=1}^k w_g P_g(s'|s,a)}_{=\overline{P}(s'|s,a)} \right) \right|$$
$$\leq \frac{6\sqrt{|S||A|}H\eta}{1-\gamma} \ .$$

Inserting these bounds and the bound from (43) into (44), we get

$$\left|\hat{\mathcal{L}}^M(d,h;i) - \mathcal{L}^M(d,h;i)\right| \leq$$
$$\frac{B(H(1+\gamma)+1)}{1-\gamma}\sqrt{\sum_{g=1}^k \frac{w_g^2}{m_g}\left(|S||A|\ln\left(\frac{2\ln(|S||A|BH/\eta)}{\eta}\right) + |S|\ln\left(1+\frac{2H}{\eta}\right) + \ln\left(\frac{2}{\delta_1}\right)\right)}$$
$$+ \frac{4BH\sqrt{|S|}\eta}{1-\gamma} + \frac{6\sqrt{|S||A|}H\eta}{1-\gamma}$$

In particular, if we use $m_{ik+g} = U_i w_g$, we get

$$\left|\hat{\mathcal{L}}^M(d,h;i) - \mathcal{L}^M(d,h;i)\right| \leq$$
$$\frac{2B(H+1)}{1-\gamma}\sqrt{\frac{1}{U_i}\left(|S||A|\ln\left(\frac{2\ln(|S||A|BH/\eta)}{\eta}\right) + |S|\ln\left(1+\frac{2H}{\eta}\right) + \ln\left(\frac{2}{\delta_1}\right)\right)}$$
$$+ \frac{4BH\sqrt{|S|}\eta}{1-\gamma} + \frac{6\sqrt{|S||A|}H\eta}{1-\gamma}$$

If we now choose $U_i \geq \frac{1}{\eta^2}\left(|A|\ln\left(\frac{2\ln(|S||A|BH/\eta)}{\eta}\right) + \ln\left(1+\frac{2H}{\eta}\right) + \frac{\ln(2/\delta_1)}{|S|}\right)$, we get

$$\left|\hat{\mathcal{L}}^M(d,h;i) - \mathcal{L}^M(d,h;i)\right| \leq \frac{6(H+1)\sqrt{|S|}(B+\sqrt{|A|})\eta}{1-\gamma} \ .$$

$\square$

We need some further definitions and then go on to show the theorem on MDRR.

**Definition 7.** *We define* $\mathrm{MR}_{\boldsymbol{w}}^k(d_i, P_{i\cdot k}, r_{i\cdot k})$ *to be the solution to* (41).

*Furthermore we define* $\widehat{\mathrm{MR}}_{\boldsymbol{w}}^k(d_i, P_{i\cdot k}, r_{i\cdot k})$ *to be the occupancy measure* $d$ *optimizing the empirical Lagrangian for MDRR, i.e.*

$$\max_d \min_h \hat{\mathcal{L}}^M(d,h,i) \ . \tag{45}$$

*After deploying $\pi_{d_i}$ for $k$ rounds, the learner updates its occupancy measure by $d_{i+1} = \widehat{\mathrm{MR}}^k_{\boldsymbol{w}}(d_i, P_{i\cdot k}, r_{i\cdot k})$.*

## F.2 FORMAL STATEMENT AND PROOF OF THEOREM 5 (MDRR)

We now show a more general version of the Theorem 5.

**Theorem 12.** *Let $d_i$ be computed by MDRR with $k \geq \frac{\ln\left(\frac{\epsilon(v-1)}{v\epsilon-1}\right) + \ln\left(\frac{5(1-\epsilon)\,\mathrm{d}_{P,r}}{\iota\delta}\right)}{\ln(1/\epsilon)}$. Suppose Assumption 1 holds and Assumption 3 holds for $k$ and parameter B. Furthermore assume that $\lambda > \max\left(6.08\xi\eta, \frac{19}{18}\xi\eta + \frac{\phi\iota}{1-\epsilon}\left(1 + \frac{1}{5.0\bar{6}\xi\eta}\right)\right)$ with $\xi$ being defined as above.*

*Further let $U_i \geq \frac{1}{\eta^2}\left(|A|\ln\left(\frac{2\ln(|S||A|BH/\eta)}{\eta}\right) + \ln\left(1 + \frac{2H}{\eta}\right) + \frac{\ln(4i^2/p)}{|S|}\right)$ be the total number of samples in round $i$, where the number of samples is given by $m_{ik+g} = w_g U_i$ with $w_g = \frac{v-1}{v^k-1}v^{g-1}$ and $v > \frac{1}{\epsilon}$. Then for any $\delta > 0$ and $p > 0$, with probability at least $1 - p$,*

$$\|d_i - d_S\|_2 \leq \delta$$

$$\text{for all } i \geq \frac{\ln(\|d_1, d_S\|_2 / \delta)}{\ln\left(1/\left(\sqrt{\frac{19\xi\eta}{18\lambda}} + \frac{1.2\phi\iota}{\lambda(1-\epsilon)}\right)\right)} + 1 .$$

*Here $\eta > 0$ and $v > \frac{1}{\epsilon}$ can be chosen arbitrarily.*

The parameter $\eta > 0$ defines a trade-off between the number of samples, number of iterations and the conditions on $\lambda$. The parameter $v > \frac{1}{\epsilon}$ defines a trade-off between the number of deployments per retraining and the required number of samples per deployment.

We first explain how Theorem 5 follows from Theorem 12. Assumption 2 ensures that $\beta \geq \alpha$, $\epsilon_p = \epsilon_r = \epsilon$, $\iota \leq \epsilon$ and $\beta \geq \alpha$. For Theorem 5, we use $\eta = \frac{\lambda}{10\xi}$ and $\lambda > \max\left(1, 3.6\frac{\beta\iota}{1-\epsilon}, \frac{1.2\beta\iota}{(1-\epsilon)\left(\frac{\epsilon+3}{4} - \sqrt{\frac{19}{180}}\right)}\right)$. We now bound $\sqrt{\frac{19\xi\eta}{18\lambda}} + \frac{1.2\phi\iota}{\lambda(1-\epsilon)}$ in order to bound the number of retrainings $i$. We see that

$$\sqrt{\frac{19\xi\eta}{18\lambda}} + \frac{1.2\beta\iota}{\lambda(1-\epsilon)} = \sqrt{\frac{19}{180}} + \frac{1.2\beta\iota}{\lambda(1-\epsilon)} < \frac{\epsilon+3}{4} .$$

where in the inequality, we use $\lambda > \frac{1.2\beta\iota}{(1-\epsilon)\left(\frac{\epsilon+3}{4} - \sqrt{\frac{19}{180}}\right)}$.

Inserting the bounds on $\lambda$ and $\eta$, we get the results described in Theorem 5.

*Proof of Theorem 12.* In general, we bound

$$\|d_{i+1} - d_S\|_2 \leq \underbrace{\left\|\widehat{\mathrm{MR}}^k_{\boldsymbol{w}}(d_i, P_{ik}, r_{ik}) - \mathrm{MR}^k_{\boldsymbol{w}}(d_i, P_{ik}, r_{ik})\right\|_2}_{T_1}$$

$$+ \underbrace{\left\|\mathrm{MR}^k_{\boldsymbol{w}}(d_i, P_{ik}, r_{ik}) - \mathrm{GD}(P_{d_i}, r_{d_i})\right\|_2}_{T_2} + \underbrace{\left\|\mathrm{GD}(P_{d_i}, r_{d_i}) - d_S\right\|_2}_{T_3}$$

where $d_S$ is some stable occupancy measure.

We begin by bounding $T_1$. For this we argue similarly to the proof of Theorem 3 in Mandal et al. [2023].

Let $\hat{h}_{i+1}$ be the dual solution to $\hat{\mathcal{L}}^M$ corresponding to $\widehat{\mathrm{MR}}^k_{\boldsymbol{w}}(d_i, P_{ik}, r_{ik})$. I.e.

$$(\widehat{\mathrm{MR}}^k_{\boldsymbol{w}}(d_i, P_{ik}, r_{ik}), \hat{h}_{i+1}) = \arg\max_d \arg\min_h \hat{\mathcal{L}}^M(d, h; i)$$

By strong duality, there has to exist a $h_{i+1}$ such that

$$(\mathrm{MR}^k_{\boldsymbol{w}}(d_i, P_{ik}, r_{ik}), h_{i+1}) = \arg\max_d \arg\min_h \mathcal{L}^M(d, h; i)$$

Using lemma 4 of Mandal et al. [2023], we can bound the $L_2$-norms of the dual solutions $\hat{h}_{i+1}$ and $h_{i+1}$ by $\frac{3|S|}{(1-\gamma)^2}$. We can thus consider the restricted set $\mathcal{H} = \left\{ h : \|h\|_2 \leq \frac{3|S|}{(1-\gamma)^2} \right\}$. Then because Assumption 3 holds, we can apply lemma 8 with $\delta_1 = p/2i^2$ and $H = 3|S|/(1-\gamma)^2$ to get,

$$\left| \hat{\mathcal{L}}^M(d_{i+1}, h_{i+1}; i) - \mathcal{L}^M(d_{i+1}, h_{i+1}; i) \right| \leq \frac{19|S|^{1.5}(B + \sqrt{|A|})\eta}{(1-\gamma)^3} \tag{46}$$

if

$$U_i \geq \frac{1}{\eta^2} \left( |A| \ln \left( \frac{2\ln(|S||A|BH/\eta)}{\eta} \right) + \ln \left( 1 + \frac{2H}{\eta} \right) + \frac{\ln \left( 4i^2/p \right)}{|S|} \right) .$$

Note that event (46) holds with probability at least $1 - \frac{p}{2i^2}$. By a union bound over all rounds, the event holds with probability at least $1 - p$ for all rounds.

The objective $\mathcal{L}^M(\cdot, h_{i+1}, i)$ is $\lambda$-strongly concave. Therefore, we have

$$\mathcal{L}^M(\widehat{\mathrm{MR}}_{\boldsymbol{w}}^k(d_i, P_{ik}, r_{ik}), h_{i+1}; i) - \mathcal{L}^M(\mathrm{MR}_{\boldsymbol{w}}^k(d_i, P_{ik}, r_{ik}), h_{i+1})$$
$$\leq -\frac{\lambda}{2} \left\| \widehat{\mathrm{MR}}_{\boldsymbol{w}}^k(d_i, P_{ik}, r_{ik}) - \mathrm{MR}_{\boldsymbol{w}}^k(d_i, P_{ik}, r_{ik}) \right\|_2^2$$

We therefore find by rearranging and using lemma 12 from Mandal et al. [2023],

$$T_1 = \left\| \widehat{\mathrm{MR}}_{\boldsymbol{w}}^k(d_i, P_{ik}, r_{ik}) - \mathrm{MR}_{\boldsymbol{w}}^k(d_i, P_{ik}, r_{ik}) \right\|_2$$
$$\leq \sqrt{\frac{2 \left( \mathcal{L}^M(\mathrm{MR}_{\boldsymbol{w}}^k(d_i, P_{ik}, r_{ik}), h_{i+1}; i) - \mathcal{L}^M(\widehat{\mathrm{MR}}_{\boldsymbol{w}}^k(d_i, P_{ik}, r_{ik}), h_{i+1}; i) \right)}{\lambda}}$$
$$\leq \frac{\sqrt{38|S|^{1.5}(B + \sqrt{|A|})\eta}}{(1-\gamma)^{1.5}} \frac{1}{\sqrt{\lambda}}$$

We now bound $T_2$ using lemma 1

$$T_2 = \left\| \mathrm{MR}_{\boldsymbol{w}}^k(d_i, P_{ik}, r_{ik}) - \mathrm{GD}(P_{d_i}, r_{d_i}) \right\|_2 = \left\| \mathrm{GD}(\overline{P}_i, \overline{r}_i) - \mathrm{GD}(P_{d_i}, r_{d_i}) \right\|_2$$
$$\leq \frac{\phi}{\lambda} \mathrm{dist}((\overline{P}_i, \overline{r}_i), (P_{d_i}, r_{d_i}))$$

with $\phi = \max(\alpha, \beta)$, with $\alpha$ and $\beta$ from Definition 4.

We can further bound this using lemma 9.

$$\frac{\phi}{\lambda} \mathrm{dist}((\overline{P}_i, \overline{r}_i), (P_{d_i}, r_{d_i})) \leq \frac{\phi}{\lambda} \frac{v^k \epsilon^{k+1}(v-1) - v\epsilon + \epsilon}{v^k(v\epsilon - 1) - v\epsilon + 1} \mathrm{dist}((P_{ik}, r_{ik}), (P_{ik+1}, r_{ik+1}))$$

We can now bound $T_3$ using lemmas 1 and 4.

$$T_3 = \|\mathrm{GD}(P_{d_i}, r_{d_i}) - d_S\|_2 \leq \frac{\phi}{\lambda} \mathrm{dist}((P_{d_i}, r_{d_i}), (P_S, r_S)) \leq \frac{\phi\iota}{\lambda(1-\epsilon)} \|d_i - d_S\|_2$$

In total we get

$$\|d_{i+1} - d_S\|_2 \leq \frac{\sqrt{38|S|^{1.5}(B + \sqrt{|A|})\eta}}{(1-\gamma)^{1.5}} \frac{1}{\sqrt{\lambda}} + \frac{\phi}{\lambda} \mathrm{dist}((\overline{P}_i, \overline{r}_i), (P_{d_i}, r_{d_i})) + \frac{\phi\iota}{\lambda(1-\epsilon)} \|d_i - d_S\|_2$$
$$\leq \frac{\sqrt{38|S|^{1.5}(B + \sqrt{|A|})\eta}}{(1-\gamma)^{1.5}} \frac{1}{\sqrt{\lambda}} + \frac{\phi}{\lambda} \frac{v^k \epsilon^{k+1}(v-1) - v\epsilon + \epsilon}{v^k(v\epsilon - 1) - v\epsilon + 1} \mathrm{dist}((P_{ik}, r_{ik}), (P_{ik+1}, r_{ik+1})) + \frac{\phi\iota}{\lambda(1-\epsilon)} \|d_i - d_S\|_2$$
$$= x_1 \delta + x_2 \|d_i - d_S\|_2$$

Where we define $x_1 = \frac{\sqrt{38|S|^{1.5}(B+\sqrt{|A|})\eta}}{(1-\gamma)^{1.5}\sqrt{\lambda}\delta} + \frac{\phi}{\lambda\delta}\frac{v^k\epsilon^{k+1}(v-1)-v\epsilon+\epsilon}{v^k(v\epsilon-1)-v\epsilon+1}\text{dist}((P_{ik},r_{ik}),(P_{ik+1},r_{ik+1}))$ and $x_2 = \frac{\phi\iota}{\lambda(1-\epsilon)}$.

We now prove that after a certain number of update iterations $i$, the occupancy measure $d_i$ is in a $\delta$ radius around a stable occupancy measure $d_S$. For this we can apply lemma 3, if we know that $x_1 + x_2 < 1$.

So we first derive criteria under which $x_1 + x_2 < 1$ holds.

From the conditions of the Theorem if follows that $v\epsilon > 1$. Using this we can derive that for any $z > 0$, if $k > \frac{\ln\left(\frac{\epsilon(v-1)}{v\epsilon-1}\right)+\ln(1/z)}{\ln(1/\epsilon)}$, then $\frac{v^k\epsilon^{k+1}(v-1)-v\epsilon+\epsilon}{v^k(v\epsilon-1)-v\epsilon+1} < z$.

We now bound

$$\frac{\phi}{\lambda\delta}\frac{v^k\epsilon^{k+1}(v-1)-v\epsilon+\epsilon}{v^k(v\epsilon-1)-v\epsilon+1}\text{dist}((P_{ik},r_{ik}),(P_{ik+1},r_{ik+1})) \leq \frac{0.2\phi\iota}{\lambda(1-\epsilon)},$$

which holds if

$$k \geq \frac{\ln\left(\frac{\epsilon(v-1)}{v\epsilon-1}\right)+\ln\left(\frac{5(1-\epsilon)\,\text{dist}((P_{ik},r_{ik}),(P_{ik+1},r_{ik+1}))}{\iota\delta}\right)}{\ln(1/\epsilon)}.$$

We then have that

$$x_1 + x_2 \leq \frac{\sqrt{38|S|^{1.5}(B+\sqrt{|A|})\eta}}{(1-\gamma)^{1.5}\sqrt{\lambda}\delta} + \frac{1.2\phi\iota}{\lambda(1-\epsilon)}.$$

Using lemma 6, with $x = \lambda$, $a = \frac{\sqrt{38|S|^{1.5}(B+\sqrt{|A|})\eta}}{(1-\gamma)^{1.5}\delta}$, $b = \frac{1.2\phi\iota}{1-\epsilon}$ and $y = 2.4$ we get that if $\lambda > \max\left(5.76a^2, a^2 + \frac{\phi\iota}{1-\epsilon}\left(1+\frac{1}{4.8a^2}\right)\right)$, then $1 > x_1 + x_2$.

We can then apply lemma 3 to see that for $i \geq \frac{\ln(\|d_1-d_S\|_2/\delta)}{\ln(1/(x_1+x_2))}+$, it holds that $\|d_i - d_S\|_2 \leq \delta$. $\qquad\square$

In the proof of Theorem 12 we use the following lemma.

**Lemma 9.** *If Assumption 1 holds with $\epsilon_p, \epsilon_r < 1$, then if $\frac{m_{ik+t}}{U_i} = \frac{v-1}{v^k-1}v^{t-1}$, it holds that*

$$\text{dist}((\overline{P}_i,\overline{r}_i),(P_{d_i},r_{d_i})) \leq \frac{v^k\epsilon^{k+1}(v-1)-v\epsilon+\epsilon}{v^k(v\epsilon-1)-v\epsilon+1}\text{dist}((P_{ik},r_{ik}),(P_{ik+1},r_{ik+1})).$$

*where $\overline{r}_i := \sum_{g=1}^{k}\frac{v^{g-1}(v-1)}{v^k-1}r_{ik+g}$ and $\overline{P}_i := \sum_{g=1}^{k}\frac{v^{g-1}(v-1)}{v^k-1}P_{ik+g}$ for some $v > 1$.*

*Proof.*

$$\text{dist}((\overline{P}_i,\overline{r}_i),(P_{d_i},r_{d_i})) \leq \sum_{g=1}^{k}\frac{v^{g-1}(v-1)}{v^k-1}(\|P_{ik+g}-P_{d_i}\|_2 + \|r_{ik+g}-r_{d_i}\|_2)$$

Note that if Assumption 1 holds with $\epsilon_p, \epsilon_r < 1$, then the map $g_d$ is contractive with unique fixed point $(P_{d_i},r_{d_i})$ and

Lipschitz coefficient $\epsilon$ (see Proposition 3). So we have for $v\epsilon \neq 1$:

$$(1 - \epsilon) \sum_{g=1}^{k} \frac{v^{g-1}(v-1)}{v^k - 1} (\|P_{ik+g} - P_{d_i}\|_2 + \|r_{ik+g} - r_{d_i}\|_2)$$

$$\leq (1 - \epsilon) \sum_{g=1}^{k} \frac{\epsilon(\epsilon v)^{g-1}(v-1)}{v^k - 1} (\|P_{ik} - P_{d_i}\|_2 + \|r_{ik} - r_{d_i}\|_2)$$

$$\leq \sum_{g=1}^{k} \frac{\epsilon(\epsilon v)^{g-1}(v-1)}{v^k - 1} (\|P_{ik} - P_{d_i}\|_2 + \|r_{ik} - r_{d_i}\|_2 - \|P_{ik+1} - P_{d_i}\|_2 - \|r_{ik+1} - r_{d_i}\|_2)$$

$$\leq \frac{\epsilon(v-1)}{v^k - 1} (\|P_{ik} - P_{ik+1}\|_2 + \|r_{ik} - r_{ik+1}\|_2) \sum_{g=1}^{k} (\epsilon v)^{g-1}$$

$$= \frac{\epsilon(v-1)(v^k \epsilon^k - 1)}{(v^k - 1)(v\epsilon - 1)} (\|P_{ik} - P_{ik+1}\|_2 + \|r_{ik} - r_{ik+1}\|_2)$$

$$= \frac{v^{k+1}\epsilon^{k+1} - v\epsilon - v^k \epsilon^{k+1} + \epsilon}{v^{k+1}\epsilon - v^k - v\epsilon + 1} (\|P_{ik} - P_{ik+1}\|_2 + \|r_{ik} - r_{ik+1}\|_2)$$

$$= \frac{v^k \epsilon^{k+1}(v-1) - v\epsilon + \epsilon}{v^k(v\epsilon - 1) - v\epsilon + 1} (\|P_{ik} - P_{ik+1}\|_2 + \|r_{ik} - r_{ik+1}\|_2)$$

$\square$