# OpenReview forum: "Performative Reinforcement Learning in Gradually Shifting Environments"
_auai.org/UAI/2024/Conference — UAI 2024 poster_

### Official Review · Reviewer_GHSh · 2024-03-19

**Q2-1 Originality-Novelty:** 2
**Q2-2 Correctness-Technical Quality:** 3
**Q2-5 Clarity Of Writing:** 3

**Q1 Summary And Contributions:**

The authors propose an extension to the performative reinforcement learning framework, where the underlying MDP is gradually changing over time instead of adapting immediately after every policy change. The authors propose a retraining algorithm that, unlike previous work, reuses historic data. The authors provide a comprehensive evaluation of retraining approaches in theory and practice.

**Q2-3 Extent To Which Claims Are Supported By Evidence:**

3: Good: the main claims are supported by convincing evidence (in the form of adequate experimental evaluation, proofs, (pseudo-)code, references, assumptions).

**Q2-4 Reproducibility:**

3: Good: key resources (e.g. proofs, code, data) are available and key details (e.g. proofs, experimental setup) are sufficiently well-described for competent researchers to confidently reproduce the main results.

**Q3 Main Strengths:**

The paper extends an interesting topic in RL to a scenario that is very plausible in practice. The authors clearly state how their contributions relate to existing literature. Table 1 provides a great overview of their theoretical results.

**Q4 Main Weakness:**

The complexity of the theoretical results seem convincing, albeit daunting for an uninitiated reader.

**Q5 Detailed Comments To The Authors:**

I am confused by the recursive formulation of Equation 2 implied by $r_{d_s^*}$ and $P_{d_s^*}$ on the right-hand side. Based on Equation 3 in [1], shouldn’t it instead be $r_{d}$ and $P_{d}$?

Equation 5 introduces a function $h(s)$ for which Equation 6 provides an update function. However, it is never explained what this function represents. Is it a state-specific Lagrange multiplier?

I would be curious to see how strongly the environment has to respond to the current policy (i.e what value of $w$) for MDRR to fail, or at least approach the performance of the baselines. Since this experiment clearly aims at a failure mode, I do not expect it to be included in the main body, but I think it would provide great insight if it was included at least in the appendix.

Typos:
- Page 2: “signifying the ~the~ compute needed to converge …”
- Page 3: “They then change to ~to~ …”
- Page 3: “On**e** common solution concept in this setting …”

[1] Performative Reinforcement Learning by Mandal et al.

**Q9 Complying With Reviewing Instructions:**

Yes

---

> ### Author Rebuttal · Authors · 2024-04-04
>
> Thank you for your valuable comments and questions. We are happy that you found the topic interesting and our extension very plausible for practical scenarios.
> # Answering Questions
> > I am confused by the recursive formulation of Equation 2 implied by $r_{d^*\_S}$ and $P_{d^*\_S}$ on the right-hand side. Based on Equation 3 in [1], shouldn't it instead be $r_{d}$ and $P_d$?
>
> Good question. Equation (3) in [1] likely contains a typo. They probably wanted to define something more akin to what we defined in Equation (2). This becomes clear when we look at Definition 3 in [1], which defines performative stability and is consistent with our definition and prior work.
>
> In particular, performative stability in the RL case defines an occupancy measure of a policy which is optimal against the environment it induces. Therefore it is a fixed point when repeatedly updating the policy. After the environment converges to $M_{d^*\_S}$, repeatedly updating the policy will result in occupancy measure $d^*\_S$ which is a best response against the environment $M_{d^*\_S}$.
>
> > Equation 5 introduces a function $h(s)$ for which Equation 6 provides an update function. However, it is never explained what this function represents. Is it a state-specific Lagrange multiplier?
>
> Thank you for pointing out that $h(s)$ is not defined in the main paper. We will change this in subsequent versions.
>
> As you correctly described, $h$ is the Lagrange multiplier of size $|S|$ and $h(s)$ is one entry in this Lagrange multiplier.
>
> > I would be curious to see how strongly the environment has to respond to the current policy (i.e what value of $w$) for MDRR to fail, or at least approach the performance of the baselines. Since this experiment clearly aims at a failure mode, I do not expect it to be included in the main body, but I think it would provide great insight if it was included at least in the appendix.
>
> Great idea! In order to evaluate this, we will do some more experiments for
> larger values of $w$. Specifically, we started experiments with $w=0.85$ and
> $w=0.95$. Right now only the experiments with $w=0.85$ are ready.
> There MDRR still converges faster, although its advantage decreases (the experiments indicate a speedup of factor $5$ against RR and DRR).
> We will report back once all results of the current experiments are ready and also include interesting findings in future versions of the manuscript.
>
> **Edit:**
> Also the experiments with $w=0.95$ are now finished.
> MDRR still converges more than $7$ times faster than RR and DRR.
> This is somewhat counterintuitive, since at such high values of $w$ the environment is almost non-stateful, and we would expect RR to have an advantage here. We believe that this phenomenon is due to the fact that MDRR uses more samples than RR and DRR and therefore has a lower variance, even at the cost of a high bias. This seems to lead to a much better convergence in the settings we studied.
>
>
> Thank you for pointing out typos.
> # Conclusion
> Thank you once again for your comments. We are happy to answer any additional questions.
> # References
> [1] Mandal, Debmalya, Stelios Triantafyllou, and Goran Radanovic. "Performative reinforcement learning." International Conference on Machine Learning. PMLR, 2023.

---

### Official Review · Reviewer_p51z · 2024-03-22

**Q2-1 Originality-Novelty:** 3
**Q2-2 Correctness-Technical Quality:** 3
**Q2-5 Clarity Of Writing:** 4

**Q10 Ethical Concerns:**

No.

**Q1 Summary And Contributions:**

This paper introduces a framework in performative reinforcement learning, focusing on how the dynamics of Markov Decision Processes evolve in response to both current and historical policy deployments. By merging insights from performative RL with environmental shifts, the paper contributes to understanding how RL agents can more realistically interact with dynamic environments. Empirical evaluations are conducted in the simulation environments to evaluate the proposed method alongside baselines.

**Q2-3 Extent To Which Claims Are Supported By Evidence:**

3: Good: the main claims are supported by convincing evidence (in the form of adequate experimental evaluation, proofs, (pseudo-)code, references, assumptions).

**Q2-4 Reproducibility:**

4: Excellent: key resources (e.g. proofs, code, data) are available and key details (e.g. proof sketches, experimental setup) are comprehensively described for competent researchers to confidently and easily reproduce the main results.

**Q3 Main Strengths:**

- It provides a detailed theoretical and experimental analysis, demonstrating the advantages of the proposed methods.
- The paper presents a novel approach by combining the ideas from performative reinforcement learning (RL) and the environment shift.
- The finite sample guarantees make the proposed method practical and relevant to real-world scenarios.
- Overall, the paper offers a thorough explanation of the methodologies and their implications and justifies the proposed approach, making it both accessible and convincing.

**Q4 Main Weakness:**

- The paper heavily relies on existing works (Mandal et al. [2023] and Brown et al. [2022]).
- Its limited analysis of non-tabular setups questions its broader practical applicability.

**Q5 Detailed Comments To The Authors:**

- Could you elaborate on the decision to focus predominantly on tabular setups?
- How do you envision addressing the challenge of extending your findings to non-tabular environments to enhance practical applicability?
- How does the proposed method of environment shift relate to reinforcement learning generalization, where RL agents learn robust policies to perform across environmental variations (such as different background colors)?

**Q9 Complying With Reviewing Instructions:**

Yes

---

> ### Author Rebuttal · Authors · 2024-04-04
>
> Thank you for your valuable comments and questions. We are happy that you find our ideas relevant and convincing.
> # Answering Questions
> > Could you elaborate on the decision to focus predominantly on tabular setups?
>
> Certainly! Unlike standard RL, where tabular settings are easy to solve, our model introduces significant complexity, making the study of tabular settings an important starting point for deeper exploration. As can be seen by our work and the work from [1], already studying tabular settings is far from trivial in RL settings with dynamically changing environments.
>
> > How do you envision addressing the challenge of extending your findings to non-tabular environments to enhance practical applicability?
>
> Great question! We note that this is an open problem even for standard performative RL [1]. We envision that we first need to solve this problem in standard performative RL and only then can extend those results to our setting.
>
> > How does the proposed method of environment shift relate to reinforcement learning generalization, where RL agents learn robust policies to perform across environmental variations (such as different background colors)?
>
> Thank you for posting this question. When trying to learn robust policies, the environment change typically does not depend on the deployed policy. In contrast to this, our framework assumes that the environment shift is dynamic and can depend on the policy. Furthermore our methods aim at finding a stable point where the policy is optimal w.r.t. the induced environment, whereas methods for RL generalization aim to make policies robust to changes in the environment.
> # Conclusion
> Thank you once again for your comments. We are happy to answer any additional questions.
> # References
> [1] Mandal, Debmalya, Stelios Triantafyllou, and Goran Radanovic. "Performative reinforcement learning." International Conference on Machine Learning. PMLR, 2023.

---

### Official Review · Reviewer_9xws · 2024-03-24

**Q2-1 Originality-Novelty:** 3
**Q2-2 Correctness-Technical Quality:** 3
**Q2-5 Clarity Of Writing:** 3

**Q1 Summary And Contributions:**

This paper studies a problem in RL that the current environment depends on the deployed policy as well as its previous dynamics, which is a generalization of performative RL. It allows to model scenarios where the environment gradually adjusts to a deployed policy. The authors adapt two algorithms from the performative prediction literature to this setting and propose a novel algorithm called Mixed Delayed Repeated Retraining (MDRR). The authors provide conditions under which these algorithms converge and compare them using three metrics: number of retrainings, approximation guarantee, and number of samples per deployment. MDRR can combine samples from multiple deployments in its training, making MDRR suitable for scenarios where the environment’s response strongly depends on its previous dynamics. Experiments show that MDRR converges significantly faster than previous approaches.

**Q2-3 Extent To Which Claims Are Supported By Evidence:**

3: Good: the main claims are supported by convincing evidence (in the form of adequate experimental evaluation, proofs, (pseudo-)code, references, assumptions).

**Q2-4 Reproducibility:**

3: Good: key resources (e.g. proofs, code, data) are available and key details (e.g. proofs, experimental setup) are sufficiently well-described for competent researchers to confidently reproduce the main results.

**Q3 Main Strengths:**

1. This paper considers a new setting that is different from and also generalizes the existing setting of performative reinforcement learning.

2. The authors study three algorithms for the proposed setting and prove their theoretical results, which are new in the area of performative reinforcement learning

3. The authors conduct experiments to verify the effectiveness of the proposed algorithms and show that  MDRR can converge much faster than other algorithms.

**Q4 Main Weakness:**

I am not an expert on this research topic. But I have questions regarding this work:

1. Since the authors mention that this work can be viewed as a generalization of the existing research on performative reinforcement learning, I think it is important to compare the theoretical results of this work and prior work such as  [Mandal et al., 2023]. Can the theoretical results in this submission be reduced to those in  [Mandal et al., 2023] once we remove the assumption that the current environment depends on its previous dynamics?

2. Why the convergence curves in Figure 1 drop suddenly in the last few steps? Can the authors provide a more detailed description of this phenomenon?

**Q5 Detailed Comments To The Authors:**

N/A

**Q9 Complying With Reviewing Instructions:**

Yes

---

> ### Author Rebuttal · Authors · 2024-04-04
>
> Thank you for your valuable comments and questions. We are happy that you find our results new and relevant.
> # Answering Questions
> > Since the authors mention that this work can be viewed as a generalization of the existing research on performative reinforcement learning, I think it is important to compare the theoretical results of this work and prior work such as [1]. Can the theoretical results in this submission be reduced to those in [1] once we remove the assumption that the current environment depends on its previous dynamics?
>
> Thank you for the great question!
> For the RR algorithm, we have some discussion on this directly below Theorems 1 and 2, which we expand in the following.
> In both the exact setting (Theorem 1 and 8) and finite-sample setting (Theorem 2 and 9), the bounds on $\lambda$ are equal to the ones in [1] when ignoring constants, $\epsilon$ factors and one $\gamma$ factor.
> The bound on the number of retrainings is also comparable in $\delta$, as all approaches require some $\ln(1/\delta)$ factor of retrainings in order to converge.
> Still, the bound on the number of retrainings can be larger in our case especially when the initial environment is far away from the one induced by a stable occupancy measure. This also holds when the current environment does not depend on its previous dynamics.
> The bounds on the number of samples are somewhat comparable, in both cases they involve a factor of  $\frac{(|A|^2 + B^2)\ln(t/p)}{\delta^4}$.
> To summarize, the bounds are not the same, also not if the environment does not depend on its previous dynamics, but in general they are similar.
>
> The DRR and MDRR algorithms use the particular structure of our setup and therefore are not studied in [1].
>
> > Why the convergence curves in Figure 1 drop suddenly in the last few steps? Can the authors provide a more detailed description of this phenomenon?
>
> Good question! Due to the stochastic nature of the setup, once the algorithms converge, the iterates $d_i$ wander around some stable occupancy measure $d_S$. We approximate $d_S$ by taking the average of the last 10 occupancy measures and call it $d_{\operatorname{last}}$. But $d_{\operatorname{last}}$ is not a perfect approximation, and in the rounds just before the last 10 rounds, the points are more likely to be close to $d_{\operatorname{last}}$ than in previous rounds.
> # Conclusion
> Thank you once again for your comments. We are happy to answer any additional questions.
> # References
> [1] Mandal, Debmalya, Stelios Triantafyllou, and Goran Radanovic. "Performative reinforcement learning." International Conference on Machine Learning. PMLR, 2023.

---

### Official Review · Reviewer_UAKp · 2024-03-24

**Q2-1 Originality-Novelty:** 2
**Q2-2 Correctness-Technical Quality:** 2
**Q2-5 Clarity Of Writing:** 2

**Q1 Summary And Contributions:**

This paper describes the problem of reinforcement learning in non-stationary environments where the changes to the environment dynamics and rewards depend on (1) the policy of the agent and (2) the previous environment dynamics (so that changes are gradual and not abrupt).  Similar problems are more generally studied in supervised learning settings, whereas only recently has reinforcement learning been considered in environments with dependency (1) but not (2).  A novel solution inspired by prior ideas from repeated retraining in both supervised and reinforcement learning settings is proposed that combines information from multiple deployments.  Theoretical results establish bounds on approximation quality, number of retrainings needed, and sample efficiency for prior solutions and the proposed algorithm.  Experimental results also demonstrate the advantages of the proposed approach.

**Q2-3 Extent To Which Claims Are Supported By Evidence:**

3: Good: the main claims are supported by convincing evidence (in the form of adequate experimental evaluation, proofs, (pseudo-)code, references, assumptions).

**Q2-4 Reproducibility:**

4: Excellent: key resources (e.g. proofs, code, data) are available and key details (e.g. proof sketches, experimental setup) are comprehensively described for competent researchers to confidently and easily reproduce the main results.

**Q3 Main Strengths:**

S1) The problem of learning in non-stationary environments is critically present in many real-world applications, and the influence of the agent's own policy on that non-stationarity is understudied.  Many such environments will have gradual changes of environment dynamics, so the problem as modeled is important, although incremental.  I expect the work to be of interest to the RL community at UAI.

S2) I appreciate the combination of both extensive theoretical results, along with some empirical validation of those theoretical results.  Having both together is not necessary for publicaation, but strengthen the overall contributions of the work.

S3) The approach, although incremental, seems like a reasonable next step after the DRR algorithm, and could it could further inspire models extended to multiagent RL.

S4) I also appreciated the inclusion of the code for the empirical study as supplementary material, which is very useful for reproducibility of the empirical results.

**Q4 Main Weakness:**

W1) The mathematical notation was inconsistent in places, which make the manuscript more difficult to follow.  This is especially surprising for a paper that places such emphasis on theoretical analysis.  Examples are provided below.

W2) I'm not sure why you chose to give less general results in the paper for the theorems than in the appendix.  I understand moving the proof details the appendix (although a quick 1-2 sentence summary of each proof would strengthen the main body of the paper).  But the more informal results do not seem more interesting or intuitive than the full results in the appendix.

This is especially important because the more informal results obscure some of the relationship between key parameters (e.g., p and \lambda in Theorem 1).

W3) Somewhat related, some of the assumptions made are divorced from the applications of the approach (especially Assumption 3).  I understand needing to make assumptions so that the proofs can be established, but it is helpful then to reconnect those assumptions with the underlying problem -- can you give any real-world examples where these assumptions would hold and would those assumptions be common?  If the assumptions are so restrictive so that they never happen in practice, then it is less clear how the theoretical results might be built upon to aid real-world RL.

W4) The experimental results would have been strengthened if they also discussed the cumulative rewards earned by each agent.  Did the improved convergence of the state occupancy actually improve rewards earned by agents?

**Q5 Detailed Comments To The Authors:**

Some examples of notational confusion include:

-- One page 3, it isn't clear why t+1 is used for the policy in your trajectory probabilities, but everything else is based on time t

-- On page 3, it is unclear which policy is the superscript and which is the subscript to the value function V.  The location of \pi and \pi' flip between Solution Concept and Definition 1.

-- The last summation in the constraints of Eq 2, 3, 4 reintroduce a, which is already a parameter to the d state occupancy

-- h in Eqs. 5-6 is never explained

-- the role of k and t might be flipped from Section 2 to Section 4?

-- It is also unclear why the notation is overloaded to replace \pi_d with d.  I'm not sure what that saves, whereas it makes it more difficult to remember when d as a parameter to a function refers to the state occupancies vs. when it is a policy

-- Sometimes subscripts and superscripts are flipped (e.g.,

Also, I would suggest not calling it an "Ablation Study" in Section 6, which only happens if you remove part of your solution (e.g., you ablate part of a DNN).  Instead, maybe call it a "Sensitivity Study" or "Robustness Study" since you are really testing the sensitivity of your solution to different hyperparameters.

**Q9 Complying With Reviewing Instructions:**

Yes

---

> ### Author Rebuttal · Authors · 2024-04-04
>
> Thank you for your valuable comments and questions. We are pleased that you find the topic important and understudied and that you are overall positive about our work.
> # Discussion of Paper Weaknesses
> ## W1) Regarding typos and notational inconsistencies
> Thank you for pointing out typos and giving valuable suggestions for improving the presentation.
>
> Indeed in the definition of the trajectory probabilities, also the policy should be indexed by $t$ and the location of $\pi$ and $\pi'$ in $V$ in Definition 1 should be flipped.
>
> In equations 5-6, $h$ is the Lagrange multiplier. We will explain this better in subsequent versions of the manuscript.
>
> The role of variable name $t$ indeed is a different one in Section 2 and Section 4/5. We now realize that this makes the manuscript harder to follow and will use a different variable name in future versions of the manuscript.
> ## W2) I'm not sure why you chose to give less general results in the paper for the theorems than in the appendix.
> Good point! We try to simplify the results in the main paper in order to make it easier for an uninitiated reader to appreciate the differences of the three algorithms. We also try to simplify the comparison to prior work.
> ## W3) Can you give any real-world examples where the theoretical assumptions would hold and would those assumptions be common?
> Great question!
> Assumption 3:
> Note that the learner can control the deployed policy and therefore ensure that the probability of choosing any action $a$ in any state $s$ is lower bounded by some positive value. This would lead to a non-zero probability of reaching any reachable state-action pair and thus ensure that Assumption 3 is satisfied. (This is e.g. achieved in $\epsilon$-greedy exploration.)
>
> In-depth analysis of Assumption 1:
> For simplicity consider a setting where only the probability transition function $P$ changes and not the reward $r$ (e.g. this holds in our experiments).
> Also consider that the response model $\mathcal{P}$ is defined in the following way:
> $$
> \mathcal{P}(d,P,r)=wP+(1-w)P^*(\pi_d)
> $$
> for some decay rate $w\in(0,1)$ and some response function $P^*(\pi_d)$.
>
> We can think of $\mathcal{P}$ being determined by a population and in each time-step a $(1-w)$ fraction of the population responds to the newly deployed policy $\pi_d$.
>
> Similar settings have been studied in performative prediction [1].
> We will assume that such settings are common in practice.
>
> We also assume that the change in $P^*$ is bounded, i.e. $|P^*(\pi_d)(s'|s,a) - P^*(\pi_{d'})(s'|s,a)|\leq c||d-d'||\_2$ for all $s,s'\in S$, $a\in A$ and some constant $c>0$. We can then derive that
> $$||\mathcal{P}(d,P,r)-\mathcal{P}(d',P',r')||_2\leq w||P-P'||_2 + (1-w)c|S|\sqrt{|A|}\cdot||d-d'||_2.$$
> If you want to see the derivation, feel free to ask.
>
> Given this equation, we choose $\epsilon_{p,p}=w$ and $\iota_p=(1-w)c|S|\sqrt{|A|}$ (all other $\iota$ and $\epsilon$ parameters are $0$). Now if $\iota_p=(1-w)c|S|\sqrt{|A|}<1$, Assumption 1 holds. That $(1-w)c|S|\sqrt{|A|}$ is smaller than $1$ is likely in many cases where $w$ is large and / or $c$ is small. The value of $w$ being large means that in each time-step, only a small fraction of the population responds to the new policy. This could likely be the case if each time-step encompasses a small amount of time. Additionally, note that the total difference $||P^*(\pi_d)-P^*(\pi_{d'})||_1$ can be in the order of $c\cdot|S|^2|A|\cdot||d-d'||_2$, so the value of $c$ might likely be small.
> ## W4) Did the improved convergence of the state occupancy actually improve rewards earned by agents?
> Good question! Figure 3 in the appendix provides some insights about this. There we see that the reward of leading agent $A_1$ in very early rounds is highest for RR and then quickly MDRR takes over. In later rounds the algorithms converge to different rewards, and none of them is a clear winner. We believe that the differences in later rounds stem from the randomization rather than from some inherent differences of the algorithms.
>
> We don't use the rewards as a metric to compare the algorithms, as this is not encoded by performative stability, the solution concept we consider in this work. The reward values are more closely measured by the solution concept of performative optimality [2,3]. There the learner tries to find a solution which gets maximum value overall. Performative optimality has so far not been investigated in RL, and it would be interesting future work to do so.
> # Conclusion
> Thank you once again for your comments. We are happy to answer any additional questions.
> # References
> 1. Ray, Mitas, et al. "Decision-dependent risk minimization in geometrically decaying dynamic environments." AAAI Conference. 2022.
>
> 2. Perdomo, Juan, et al. "Performative prediction." International Conference on Machine Learning. 2020.
>
> 3. Mandal, Debmalya, Stelios Triantafyllou, and Goran Radanovic. "Performative reinforcement learning." International Conference on Machine Learning. 2023.

---

### Meta-Review · Area_Chair_oGz6 · 2024-04-17

The paper presents a novel framework in performative RL that can deal with non-stationary environments, including the case in which the environment might change because of the policy. As mentioned by the reviewers, this is an important step towards more realistic RL in dynamic environments. The reviewers have also appreciated the interesting theoretical contributions, as well as the sound empirical evaluation. A recurring but minor concern was the potentially incremental novelty with respect to [Mandal et al 2023]. In the discussion, the authors clarified some points, especially related to how realistic the assumptions were in this context. After the discussion, all reviewers agree that this is an interesting paper and they recommend accepting it to UAI.